# High-fidelity parallel entangling gates on a neutral-atom quantum computer

Simon J. Evered[1,7], Dolev Bluvstein[1,7], Marcin Kalinowski[1,7], Sepehr Ebadi[1], Tom Manovitz[1], Hengyun Zhou[1,2], Sophie H. Li[1], Alexandra A. Geim[1], Tout T. Wang[1], Nishad Maskara[1], Harry Levine[1,6], Giulia Semeghini[3], Markus Greiner[1], Vladan Vuletić[4,5] & Mikhail D. Lukin[1✉]

The ability to perform entangling quantum operations with low error rates in a scalable fashion is a central element of useful quantum information processing[1]. Neutral-atom arrays have recently emerged as a promising quantum computing platform, featuring coherent control over hundreds of qubits[2,3] and any-to-any gate connectivity in a flexible, dynamically reconfigurable architecture[4]. The main outstanding challenge has been to reduce errors in entangling operations mediated through Rydberg interactions[5]. Here we report the realization of two-qubit entangling gates with 99.5% fidelity on up to 60 atoms in parallel, surpassing the surface-code threshold for error correction[6,7]. Our method uses fast, single-pulse gates based on optimal control[8], atomic dark states to reduce scattering[9] and improvements to Rydberg excitation and atom cooling. We benchmark fidelity using several methods based on repeated gate applications[10,11], characterize the physical error sources and outline future improvements. Finally, we generalize our method to design entangling gates involving a higher number of qubits, which we demonstrate by realizing low-error three-qubit gates[12,13]. By enabling high-fidelity operation in a scalable, highly connected system, these advances lay the groundwork for large-scale implementation of quantum algorithms[14], error-corrected circuits[7] and digital simulations[15].

Errors limit the computational capabilities of current quantum devices and must be made sufficiently low to permit efficient quantum error correction. In particular, two-qubit-gate error rates below 1% (that is, fidelities above 99%) are required to surpass quantum error-correcting thresholds[7]. Moreover, maintaining a combination of such low error rates, highly parallel control and a high degree of connectivity while scaling system size is crucial to realizing large-scale quantum computers. Although high-fidelity entangling operations were realized on isolated qubit pairs early on[16–19], only recently have these techniques been extended to larger systems. State-of-the-art examples include 99.4% fidelity on a 72-qubit superconducting chip[20] and 99.4–99.6% fidelity on a 31-ion chain[21]. Scaling these systems to even larger numbers of qubits while maintaining low error and efficient control is an exciting frontier[22,23], yet it also presents substantial platform-specific scientific and engineering challenges.

Recently, arrays of neutral atoms have emerged as a promising quantum processing platform capable of coherent control of hundreds of qubits[2,3] for analogue quantum simulations. This platform also features a flexible, dynamically reconfigurable architecture[4], whereby entangling operations can be performed between neutral-atom qubits with arbitrary connectivity and in a highly parallel manner. Although these capabilities open unique opportunities for both large-scale digital simulations[1] and computation with error-corrected qubits[7], the main outstanding challenge in the field has been to improve the two-qubit

gate fidelity substantially above the previously demonstrated value of approximately 97.5% (refs. 5,24). In this article, we experimentally realize two-qubit controlled phase (CZ) gates with 99.5% fidelity while operating on up to 60 neutral-atom qubits in parallel, closing the gate-fidelity gap to other state-of-the-art platforms[20,21,23,25]. This advance is achieved by using a family of optimal gate schemes[8,26] relying on the Rydberg-blockade mechanism that are robust to experimental imperfections and spontaneous scattering, alongside the implementation of several experimental tools to overcome previously dominant error sources. To characterize the two-qubit gates, we use several complementary benchmarking methods using repeated gate applications, each giving consistent results. Finally, these techniques are generalized to entangling operations involving a higher number of qubits, allowing us to experimentally realize parallel, high-fidelity, three-qubit entangling gates.

## Neutral-atom entangling gates

In our approach, quantum information is encoded in long-lived $m_F = 0$ hyperfine qubits[27], in which high-fidelity (>99.97%) coherent single-qubit rotations are driven by Raman laser pulses. Entangling operations are performed in parallel by positioning the atoms, trapped in individual optical tweezers, into designated gate sites, followed by state-selective excitation into highly excited atomic

[1]Department of Physics, Harvard University, Cambridge, MA, USA. [2]QuEra Computing Inc., Boston, MA, USA. [3]Harvard John A. Paulson School of Engineering and Applied Sciences, Harvard University, Cambridge, MA, USA. [4]Department of Physics, Massachusetts Institute of Technology, Cambridge, MA, USA. [5]Research Laboratory of Electronics, Massachusetts Institute of Technology, Cambridge, MA, USA. [6]Present address: AWS Center for Quantum Computing, Pasadena, CA, USA. [7]These authors contributed equally: Simon J. Evered, Dolev Bluvstein, Marcin Kalinowski. ✉e-mail: lukin@physics.harvard.edu

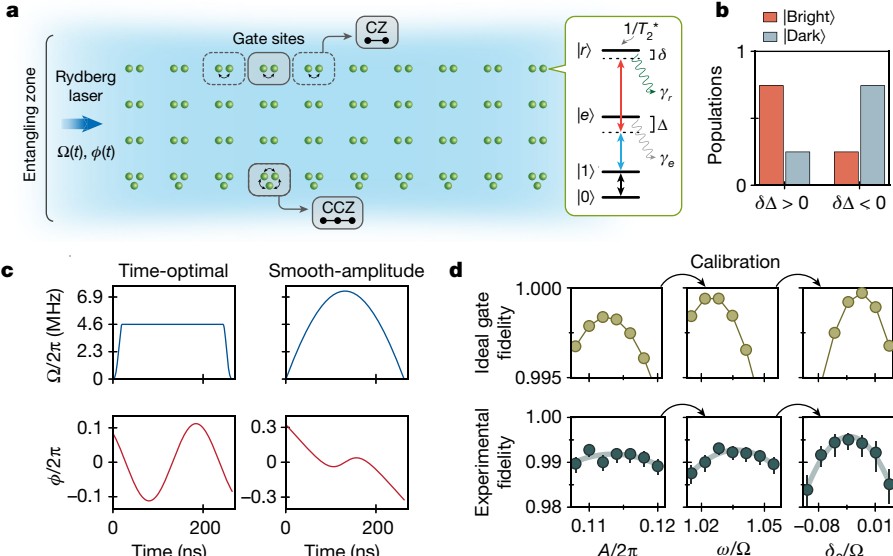

**Fig. 1 | Parallel implementation of high-fidelity entangling gates on a neutral-atom quantum computer. a**, Entangling gates are implemented by arranging atoms into designated gate sites in which they interact by means of Rydberg-blockade interactions when pulsing global Rydberg lasers. Two-qubit or three-qubit gates are performed by modulating the Rabi frequency $\Omega(t)$ and phase $\phi(t)$ profiles of the laser driving the first leg of the two-photon Rydberg excitation. Inset, atoms in the qubit state $|1\rangle$ are excited to the Rydberg state $|r\rangle$ through an intermediate excited state $|e\rangle$, whereas atoms in $|0\rangle$ are not excited. The main gate error sources include Rydberg-state decay $\gamma_r$, intermediate-state scattering $\gamma_e$ and Rydberg dephasing $T_2^*$. **b**, Numerical comparison of average bright and dark state populations during the Rydberg gate. Choosing opposite

intermediate-state ($\Delta$) and two-photon ($\delta$) detuning signs at the beginning of the gate maximizes population in the dark state, which minimizes the intermediate-state scattering error. **c**, Entangling gates are implemented with a single Rydberg laser pulse with smooth phase modulation $\phi(t)$, whose slope corresponds to a two-photon detuning $\delta(t)$. Several global parameters characterize the gate, allowing for a family of possible gate implementations, including a parameterized version of the time-optimal CZ gate[8] and a smooth-amplitude CZ gate. **d**, Example gate calibration sequence. Tuning individual parameters of the parameterized time-optimal gate phase profile $\phi(t) = A\cos(\omega t - \varphi) + \delta_0 t$ allows for fast and simple global calibration (see Extended Data Fig. 5 for further experimental data). Error bars represent 68% confidence intervals.

Rydberg states using a two-photon transition (Fig. 1a). Errors in such quantum operations can occur owing to spontaneous emission from the intermediate atomic state $|e\rangle$, atomic temperature effects, Rydberg-state decay during the gate (Fig. 1a inset), as well as miscalibrations and experimental imperfections, such as laser noise or inhomogeneity.

We address these errors through the combination of gate schemes relying on optimal control and experimental improvements. Our method for gate implementation is inspired by the recently proposed time-optimal gate by Jandura and Pupillo[8], which uses a numerically optimized continuous phase profile[8,24,28] for a single laser pulse (as opposed to a discrete phase jump between two laser pulses[5]). We generalize this gate scheme to a family of single-pulse gates with a small set of tunable gate parameters, including a version of the time-optimal gate consisting of a parameterized sinusoidal phase modulation, as well as a second, smooth-amplitude gate (Fig. 1c; see Methods for details).

We calibrate these gates by tuning several global parameters (Fig. 1d), which lends robustness to experimental imperfections: an optimal set of gate parameters can be found even in the presence of systematic offsets, such as finite Rydberg laser pulse rise time (Extended Data Fig. 4a). We further optimize our control pulses to suppress scattering from the short-lived intermediate state $|e\rangle$ by minimizing population in the 'bright' dressed state ($|B\rangle \propto |1\rangle + \sqrt{\frac{2\Omega}{\Delta}}|e\rangle + |r\rangle$) containing $|e\rangle$ and maximizing population in the 'dark' state ($|D\rangle \propto -|1\rangle + |r\rangle$) not containing $|e\rangle$ (in which $\Omega$ is the two-photon Rabi frequency and $\Delta$ is the intermediate-state detuning)[9,29]. This optimization is achieved through the appropriate selection of the relative signs of the intermediate and two-photon detunings (Fig. 1b), as well as through smooth pulse shaping for the smooth-amplitude gate (Extended Data Fig. 2).

Our experimental realization makes use of the apparatus described previously in refs. 2,4, with which we rearrange [87]Rb atoms into

programmable, defect-free arrays. Two main experimental upgrades facilitate high-fidelity entangling-gate operation. First, we suppress scattering by substantially increasing intermediate-state detuning while maintaining a high two-photon Rabi frequency ($\Omega/2\pi = 4.6$ MHz), enabled by excitation to a lower-lying ($n = 53$) Rydberg state with a tenfold higher power laser (Extended Data Fig. 1). Second, to suppress decoherence from atomic velocity and position fluctuations, we implement $\Lambda$-enhanced grey molasses cooling and an improved optical pumping technique (Methods) to achieve colder temperatures (radial phonon occupation $\bar{n} \approx 1-2$).

## Entangling-gate characterization

To characterize the CZ gates realized with this approach, we create arrays of ten Bell pairs by arranging qubit pairs into separated gate sites (Fig. 2a) and pulsing global Rydberg and Raman lasers. Using the parameterized time-optimal gate from Fig. 1c, we create a Bell state $|\Phi^+\rangle = \frac{1}{\sqrt{2}}(|00\rangle + |11\rangle)$ (ref. 5), which is then characterized by measuring the populations of $|00\rangle$ and $|11\rangle$ and the oscillation amplitude of the two-atom parity $\langle \sigma_1^z \sigma_2^z \rangle$ on applying a global single-qubit $\pi/2$ pulse of variable phase (Fig. 2b). We extract a raw Bell-state fidelity of 98.0(2)%, exceeding previous work by about 2% (ref. 5), already suggesting a greatly improved gate fidelity. Because this Bell-state fidelity seems to be dominated by state preparation and measurement (SPAM) errors (Methods), to characterize the fidelity of the entangling gate more systematically, we apply an odd-numbered train of CZ gates to repeatedly entangle and disentangle the pairs and then characterize the fidelity of the final resulting Bell state[25,30] (Fig. 2c). We fit the decreasing fidelity to an exponential decay to extract a CZ gate fidelity $F_{CZ} = 99.52(4)\%$ (Fig. 2d).

As a separate characterization of the gate fidelity, we apply random global single-qubit rotations between sequences of CZ entangling

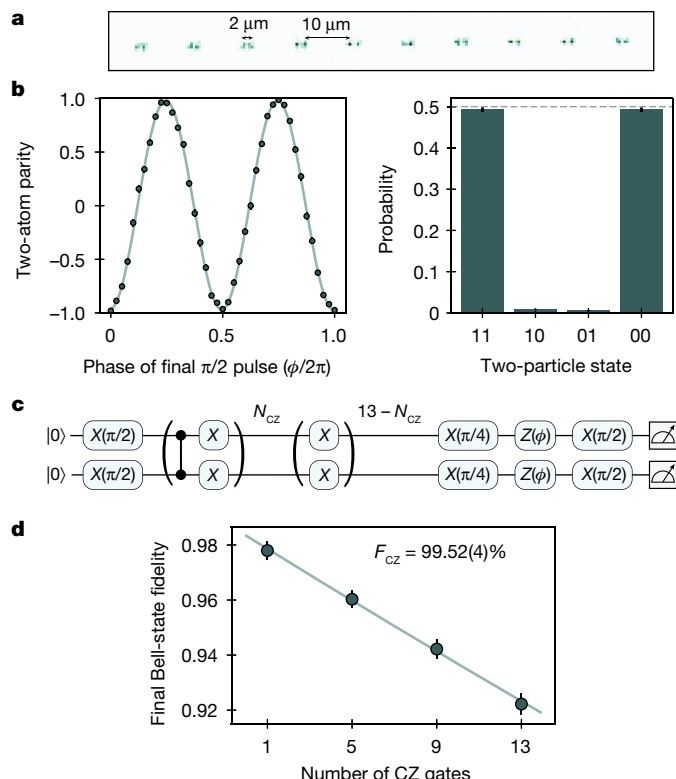

**Fig. 2 | High-fidelity CZ gates characterized with Bell states. a**, Parameterized time-optimal CZ gates are implemented on 20 atoms in parallel. **b**, Raw Bell-state measurements on application of a single CZ gate, with raw Bell-state fidelity of 98.0(2)%. We estimate a SPAM-corrected fidelity of 99.4(4)% (not plotted; see Methods for details). **c**, Circuit used to benchmark two-qubit gate fidelity by making a Bell state after an odd number of CZ gates interleaved with single-qubit $X$ gates. **d**, Decay of Bell-state fidelity after applying a variable number of CZ gates. A fidelity of 99.52(4)% per CZ gate is extracted from an exponential fit. The $y$ axis is rescaled by a SPAM correction factor that does not affect the extracted gate fidelity (Methods). Note that the same number of single-qubit gates is performed for each point, so that no normalization of single-qubit errors is necessary, which thereby results in a reduced $y$-axis intercept. Error bars represent 68% confidence intervals.

gates (Fig. 3a). This method averages over different states involved in the entangling operation similar to randomized benchmarking (see Extended Data Fig. 3 for numerical comparisons)[10,11,31,32]. In the absence of errors during the gate sequence, a precisely calculated final single-qubit operation returns the qubit pair to their initial $|00\rangle$ state. Applying a sequence of up to 20 CZ gates with random single-qubit rotations in between, we fit the decaying state fidelity as a function of CZ gate number and extract $F_{CZ} = 99.54(2)\%$ (Fig. 3b), consistent with the Bell-state method of Fig. 2d. Not only do these methods agree quantitatively but, in practice, we optimize the gate with this global randomized benchmarking method (Extended Data Fig. 5) and find that the exact same parameters are optimal for generating Bell states. The qubits also acquire a single-particle phase during the CZ gate, which this benchmarking approach eliminates by using $X$ gates in between CZ gate pairs (Fig. 3a). Therefore, we use a second method of global randomized benchmarking without these $X$ gates, which allows for calibration of the single-particle phase (used for calibrating the Bell-state measurement in Fig. 2b) and also benchmarks a gate fidelity of $F_{CZ} = 99.48(2)\%$ (Extended Data Fig. 6).

We next demonstrate that our gate methods are versatile, for which various pulse profiles can all realize a high-fidelity CZ gate. Specifically, in Fig. 3b, we also realize and benchmark the smooth-amplitude gate

(Fig. 1c) and achieve a similar fidelity of $F_{CZ} = 99.55(3)\%$. Different gate implementations can be tailored to specific use cases; for example, the smooth-amplitude gate strongly suppresses scattering even with a closer-detuned excitation, which can help achieve high gate fidelities in situations in which laser intensity is limited.

## Scaling up

We next explore the scalability of our approach to larger system sizes. Despite the fact that all calibration and control is done globally and not for individual gate sites, we find that the fidelity of the time-optimal gate is constant across the ten individual gate sites within statistical error (Fig. 3c). This observation of homogeneity across the array highlights the inherent potential for scalability: more gate sites do not increase the calibration overhead. Motivated by this observation, in Fig. 3d, we extend to a 60-qubit system by using larger Rydberg beams (while maintaining the same intensity) and achieve a gate fidelity of $F_{CZ} = 99.48(2)\%$ with good homogeneity across the array (Fig. 3e).

To understand the requirements for continued scaling and realizing high-fidelity operation in even larger system sizes, we analyse the physical error sources in the system. In particular, we compare observed gate fidelities to detailed modelling, which uses two eight-level atomic systems (Extended Data Fig. 3a) with quantitative decoherence rates informed by experimental measurements. This modelling accounts for the remaining CZ gate infidelity and reveals four main error sources (Extended Data Table 1): Rydberg decay, coupling to the other Rydberg $m_J$ level, intermediate-state scattering and our measured ground-Rydberg $T_2^* = 3$ µs, which is dominated by laser-light-shift fluctuations and finite atomic temperature. We further analyse error sources by studying gate–site correlations in the experimental data. We observe that high-weight correlated errors are largely absent from our data, which suggests the feasibility of stable, large-scale operation (Extended Data Fig. 7). Careful analysis reveals small growth in the covariance between neighbouring gate sites (Extended Data Fig. 8), which can result from either correlated detuning fluctuations (corresponding to our $T_2^*$) or long-range interactions caused by a Rydberg atom decaying into an adjacent Rydberg state (corresponding to our finite Rydberg lifetime; see further discussion in Methods).

Informed by these microscopic error sources, we conclude that the dominant challenge in maintaining high fidelity at an even larger number of parallel gate sites is to continually scale laser power and maintain beam homogeneity, as the other decoherence mechanisms seem to be independent of system size. However, we emphasize that, even with the present laser parameters, system sizes can be directly increased to hundreds of qubits by shuttling atoms in and out of the entangling zone during a quantum circuit[4,33] or by redirecting the beam to dynamically redefine the position of the entangling zone.

## Fast multi-qubit gates

Finally, we explore the generalization of these methods to multi-qubit gates. Using optimal control methods, we find a time-optimal CCZ gate and corresponding ansatz phase profile (Fig. 4c) that realizes a native CCZ gate between three qubits[5,12,13] in a time only 44% longer than the time-optimal CZ and faster than other known CCZ profiles[8]. This CCZ gate is realized experimentally by rearranging triplets of atoms into triangular gate sites (Fig. 4a) and applying the CCZ pulse profile with our global laser pulse. We characterize our CCZ gate using the sequence described in Fig. 4b to repeatedly entangle and disentangle a three-qubit Greenberger–Horne–Zeilinger (GHZ) state and subsequently measure the final GHZ-state fidelity (see Extended Data Fig. 9d for example GHZ states). Although this approach does not constitute a rigorous benchmarking of the CCZ gate fidelity (which can be done using randomized benchmarking[13] for example), our data indicate

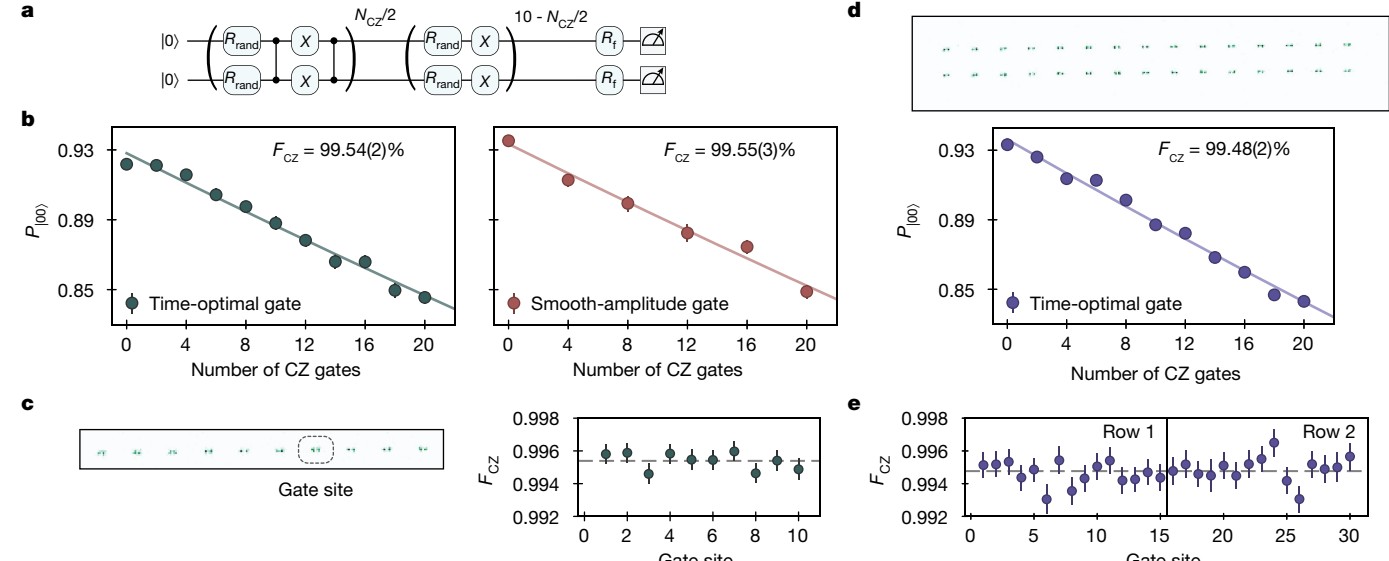

**Fig. 3 | Global randomized benchmarking of the CZ gate. a**, Circuit used to benchmark two-qubit gate fidelity with global randomized benchmarking, in which random rotations $R_{rand}$ are sampled from a Haar-random distribution (Methods) and a final rotation $R_f$ is precomputed to return population to the initial product state $|00\rangle$ in the absence of gate errors. All 21 single-qubit rotations are applied for each data point, such that the number of single-qubit operations is independent of the number of two-qubit gates. **b**, Benchmarking CZ gates on 20 atoms in parallel. We plot the probability of return to the initial product state as a function of the number of applied CZ gates for both the parameterized time-optimal gate and the smooth-amplitude gate, extracting fidelities of 99.54(2)% and 99.55(3)%, respectively. **c**, The extracted time-optimal gate fidelity is consistent within statistical error across the ten gate sites. **d**, Scaling to larger systems, we measure a comparable fidelity of 99.48(2)% for the time-optimal gate implemented on 60 qubits in parallel. **e**, Site-by-site analysis reveals homogeneous, high-fidelity gate performance across the two rows.

high-performing three-qubit entangling gates across 21 qubits in parallel, consistent with a fidelity $F = 97.9(2)\%$ (Fig. 4d). These optimal control methods extend to higher-qubit-number controlled-Z gates. We numerically search for and find fast gates for up to six qubits (Fig. 4e), with gate times markedly shorter than those required to decompose an $N$-qubit controlled-Z gate into $2N$ CZ gates and various single-qubit

gates[34]. Generically, with these global pulses and Rydberg blockade, one can natively realize symmetric, diagonal gates[8] (for example, CPHASE gates as illustrated in Extended Data Fig. 4b), which are important for efficient realization of digital quantum simulation algorithms[15].

## Discussion and outlook

Our results enable a new era of high-fidelity digital circuits with neutral atoms. On the basis of the detailed microscopic understanding of error sources, we anticipate various paths to further improve gate fidelity in future work. For example, performing the gate at three times higher Rabi frequency and two times further detuning would theoretically result in a gate fidelity of 99.9%. This would require suppressing the coupling to the adjacent Rydberg state, optimization of pulse rise times and managing high laser intensity[35] (Methods). The understanding of microscopic error sources can also be used to analyse the type of error, that is, the decomposition into different Pauli channels, atom loss and leakage[36], as described in Extended Data Table 1.

These observations open the door for explorations of large-scale quantum error correction with efficient parallel control of logical qubits[4,7]. The remaining ingredients associated with mid-circuit readout can be implemented by moving atoms[33] away from the entangling zone to a readout zone, using a second atomic species as ancilla qubits[37], shelving data qubits in auxiliary atomic levels[38], non-destructive readout with optical cavities[39,40] or ancillary atomic ensembles[41]. Alkaline-earth atoms also present further opportunities, including single-photon Rydberg excitation and nuclear spin control[42–44], as well as using erasure conversion for efficient quantum error correction[45–47]. Furthermore, high-fidelity multi-qubit gates enable many possible scientific directions based on digital quantum simulation[27] of models including non-Abelian topological physics[15], quantum gravity[48,49] and quantum chemistry[14]. Combining the analogue capabilities of the neutral-atom platform with digital circuits[4] opens the door for hybrid analogue–digital quantum simulation, including techniques such as shadow tomography[50]. Finally, the

**Fig. 4 | Realizing fast CCZ gates. a**, Entangling zone comprising three-qubit gate sites, with 21 qubits in total. The triangular configuration enables strong, symmetric interactions between three qubits. **b**, Circuit used to calibrate the CCZ gate, by generating GHZ states $(|000\rangle + |111\rangle)/\sqrt{2}$ after applying two, six and ten CCZ gates. **c**, Phase profile used for implementing a parameterized CCZ gate with a single, fixed-amplitude pulse. **d**, Raw GHZ-state fidelity as a function of CCZ gates applied, with an exponential decay fit $\propto 0.979(2)^{N_{CCZ}}$ (see Extended Data Fig. 9d for example GHZ-state characterization). **e**, Theoretical scaling of gate duration as a function of the number of qubits for native Rydberg-blockade multi-qubit gates (see Extended Data Fig. 9b for corresponding phase profiles).

high-fidelity gate can be used as a tool for other applications with neutral atoms, for example, creating a wide variety of entangled states for use in metrology[51,52], and enabling new optical lattice simulators[53–56].

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

# Methods

## Experimental system

We stochastically load hundreds of $^{87}$Rb atoms into a programmable array of 850-nm optical tweezers generated by a spatial light modulator. A second set of 810-nm optical tweezers generated by a crossed pair of acousto-optic deflectors is then used to rearrange into defect-free arrangements[2]. Subsequently, the atoms are cooled first with polarization gradient cooling on the 780-nm D2 line and then with Λ-enhanced grey molasses cooling on the 795-nm D1 line[57–60]. We implement Λ-enhanced grey molasses cooling owing to experimental simplicity (simply combined with our existing polarization gradient cooling path) as well as the potential for enhanced loading[58] (which we do not use in this work as it reduces our cycle rate). We estimate an average radial motional quantum number of $\bar{n} \approx 1 - 2$ from fitting a drop-recapture curve[61] to about 10 μK in approximately 1-mK-deep traps.

Following rearrangement and cooling, we prepare atoms in the qubit basis, formed from clock states $|0\rangle = |F = 1, m_F = 0\rangle$ and $|1\rangle = |F = 2, m_F = 0\rangle$. In our previous work[4,5], we pump into $|F = 1, m_F = 0\rangle$ using Raman-assisted optical pumping, in which we repeatedly apply π pulses on the $|F = 1, m_F = -1\rangle \rightarrow |F = 2, m_F = -1\rangle$ transition and the $|F = 1, m_F = +1\rangle \rightarrow |F = 2, m_F = +1\rangle$ transition, followed by resonant depumping of the $|F = 2\rangle$ manifold. The main disadvantage of this scheme is that it can require scattering many photons (we performed 40–70 pumping cycles in refs. 4,5) to end up in $|F = 1, m_F = 0\rangle$ and, empirically, we find that this causes enough heating to negate the benefits of the colder atoms following the Λ-enhanced grey molasses cooling step we use in this work.

To address this challenge, here instead we first optically pump into the $|F = 2, m_F = +2\rangle$ stretched state with $\sigma^+$-polarized 780-nm light, which scatters only several photons[38,62]. Then we rotate the magnetic field by 90° such that the Raman laser propagation axis has an orthogonal component and can thus drive $\sigma^{\pm}$-polarized transitions in the hyperfine manifold (see ref. 63). We apply two separate Raman pulses that transfer population first from $|F = 2, m_F = +2\rangle$ to $|F = 1, m_F = +1\rangle$ and then to $|F = 2, m_F = 0\rangle$. We use Knill composite π pulses[64] for these transfer steps and suppress unwanted transitions to adjacent $m_F$ states by using Gaussian-shaped optical pulses[64]. We note that, during this transfer process, the magnetic field direction is along the axis of the tweezer, which quadratically suppresses vector light shifts from the tweezer polarization gradient[65] that would otherwise limit coherence of the non-clock $m_F$ states. Finally, we rotate the magnetic field back to its original configuration before performing quantum circuits.

The measurements in Fig. 2b and Fig. 4d have better SPAM performance than other measurements in this paper owing to suppression of previously undetected resonant leaked light that we discovered at the end of our measurements, as well as adding a final eight rounds of our previous Raman-assisted optical pumping (also using Gaussian-shaped optical pulses to suppress off-resonant excitation) to further improve the state preparation fidelity. With leaked light managed and with these two steps of optical pumping, we achieve both low temperatures and an estimated pumping fidelity of about 99.7–99.8%, probably dominated by residual leaked light.

To measure in the computational basis, we illuminate with a strong resonant light coupling $F = 2$ to the $F' = 3$ on the D2 transition, which heats up and expels all atoms in the $|1\rangle$ state; the remaining atoms are imaged in $|0\rangle$. We estimate the combined fidelity of pushout and imaging to be about 99.83%. Finally, we also have a background atomic loss of roughly 0.25% before the circuit begins and about 0.1% after the circuit ends, dominated by a 10-s vacuum lifetime.

To drive arbitrary single-qubit rotations, we use a Raman laser system[63], which globally illuminates the atoms with a Raman Rabi frequency of 1 MHz. Single-qubit rotations are implemented using robust BB1 pulses[64,66], whereas $Z$ rotations are implemented by adjusting the phase in the control software. By applying sequences of random single-qubit rotations, we estimate a fidelity of 99.97% when sampling over a Haar-random ensemble of single-qubit rotations, implemented by a combination of two $Z$ rotations and an arbitrary BB1 pulse. This fidelity of 99.97% is consistent with the Raman scattering limit at our 180-GHz detuning and therefore fidelity can be improved by detuning the Raman laser further.

## Rydberg excitation

Extended Data Fig. 1 presents an overview of the atomic-level structure used and an example pulse sequence for running a quantum circuit, largely the same as our previous work[4]. The atoms are excited from the $|F = 2, m_F = 0\rangle$ state to the $53S_{1/2}$ Rydberg state in a two-photon scheme with a 420-nm $\sigma^+$-polarized and a 1,013-nm $\sigma^-$-polarized light. We are able to increase the intermediate-state detuning from our previous works, while maintaining similar or higher two-photon Rabi frequencies, in two ways. First, by operating at $n = 53$, we benefit from a 50% increase in the Rabi frequency compared with the previously used $n = 70$. Furthermore, we upgraded our 10-W 1,013-nm fibre amplifier laser to a 100-W 1,013-nm laser (IPG Photonics), which we operate at 20–50 W with a duty cycle <1%. Combined, this allows us to operate at the intermediate-state detuning of $\Delta/2\pi = 7.8$ GHz with a Rabi frequency of $\Omega/2\pi = 4.6$ MHz. For the data in Fig. 3d,e, we use the same Rabi frequency but an intermediate-state detuning of 6.3 GHz, which marginally increases scattering but allows us to work with lower Rydberg beam intensity.

The 1,013-nm laser seed originates from a TOPTICA DL pro external-cavity diode laser, which is then locked to and filtered through a Stable Laser Systems ultra-low-expansion cavity (finesse of 50,000 at 1,013 nm) that then injection locks another laser diode, which then seeds our 100-W laser. Our 420-nm laser is an 840-nm TiSapph (M Squared), which is locked (but not filtered) to the same cavity (finesse of 30,000 at 840 nm) and then frequency doubled with an M Squared ECD-X. For realizing the pulses and waveform shaping for quantum gates, we use an arbitrary waveform generator (Spectrum M4i.6631-x8) that allows for arbitrary amplitude, frequency and phase control, which simultaneously drives two 420-nm acousto-optic modulators (AOMs) (MQ240-B40A0,2-UV, AA Opto Electronic) in a tandem configuration and maps the RF waveform onto the 420-nm light. The 1,013-nm AOM (M1377-aQ80L-1 coated for 1,013 nm, Isomet) is pulsed on for several hundred microseconds and intensity stabilized during the duration of the entire circuit.

The two Rydberg beams are shaped into flat intensity top-hat profiles by spatial light modulators to maximize intensity while maintaining homogeneity across the gate region[2]. For the 20-atom, one-row data, we aim for a flat 10 μm × 10 μm beam cross-section (to suppress sensitivity to drift) and for the 60-atom, two-row data, we aim for a flat 20 μm × 10 μm region so that both rows are homogeneous. We tune beam parameters, including $X$ and $Y$ positions, focus and aberrations, to optimize homogeneity as measured by the differential light shift on the hyperfine qubit states. We stabilize the beam positions using a reference camera and motorized mirror mounts. To compensate for relative drift between the beam position and the atom array, we recalibrate the position often (several times per day) by stepping the beam positions to maximize the intensity at the atoms as measured by the differential light shift on the qubit transition, which takes about 5 min. We find that keeping the beams well centred on the atoms is important to ensure homogeneity and reduce sensitivity to relative beam drifts, and further find that gate parameters are highly reproducible (consistently reproducing fidelities of 99.5%) as long as the beams are properly positioned.

We extract the single-photon Rabi frequencies of the Rydberg excitation using measurements of the two-photon Rabi frequency and the 420-nm light shift on the ground-Rydberg transition, taking note of the appropriate Clebsch–Gordan coefficients for the multiple intermediate states. We extract $\Omega_{420} = 2\pi \times 237$ MHz and $\Omega_{1013} = 2\pi \times 303$ MHz,

in which we adopt a convention such that the two-photon Rabi frequency is given by $\Omega = \Omega_{420}\Omega_{1013}/2\Delta$. Note that, because of the presence of several intermediate states, the first-order scattering estimate is proportional to $\frac{4}{3} \times (\Omega_{420}/2\Delta)^2$ and thus—from the scattering perspective—the effective single-photon Rabi frequencies are well balanced (273 MHz versus 303 MHz).

Finally, we comment on the motivation for choosing the principal quantum number $n = 53$ for the Rydberg state. There are several effects that depend on $n$, including the finite Rydberg-state lifetime ($\propto n^3$ for radiative decay, $\propto n^2$ for black-body decay), matrix elements influencing the 1,013-nm Rabi frequency ($\propto n^{-3/2}$), interaction energy ($\propto n^{11}$) and sensitivity to electric fields ($\propto n^7$). Weighing these relative benefits, we work at $n = 53$, for which Rydberg lifetime effects begin to become more relevant but for which the matrix element is favourable for increasing intermediate-state detuning. A technical challenge relevant to this choice is that, because of the small blockade radius, we place atoms at 2-μm separation to achieve strong interaction strength $V_{Ryd}/2\pi \approx 450$ MHz. Operating at such close spacing is enabled by using high-numerical-aperture objectives (NA = 0.65 from Special Optics) and allows us to pack many atoms into the entangling zone.

## Parameterized entangling gates

Inspired by the methods used in ref. 8 to find the time-optimal gates, we use optimal control to design gates with phase profiles given by simple analytical formulas. We find that this approach makes the gate experimentally robust and reproducible, as small systematic offsets (for example, rise time, atom–atom separation etc.) can often be compensated for and are captured by a slightly different set of optimal parameters.

In this work, we focus on two main gates: one with a fixed amplitude and a phase profile similar to the time-optimal gate of ref. 8 and a second one in which the amplitude is also varied. We note that, in the past, many schemes were proposed to implement two-qubit gates with Rydberg interactions[5,8,29,35,67–75], to engineer robustness to errors[28,76–84] and to perform multi-qubit gates[85–91]. The fixed-amplitude gate, which we refer to as 'time-optimal' because it resembles and is only 0.2% slower than the Jandura–Pupillo gate[8], has the phase profile given by

$$\phi(t) = A\cos(\omega t - \varphi_0) + \delta_0 t. \qquad (1)$$

This profile is plotted in the left column of Fig. 1c for the parameters

$$A = 2\pi \times 0.1122, \quad \omega = 1.0431\,\Omega,$$
$$\varphi_0 = -0.7318, \qquad \delta_0 = 0 \times \Omega,$$

which constitutes an exact gate with time $(\Omega T/2\pi) = 1.215$, that is, slightly longer than a resonant single-atom $2\pi$ pulse. Note that this set of parameters is not unique and other parameter values can also realize an exact gate, for example, at non-zero detuning $\delta_0$ as used in Extended Data Fig. 4a.

The smooth-amplitude gate has a varying phase and a varying Rabi frequency of the 420-nm laser. We used optimal control methods in a three-level atomic system to find a gate that optimally suppresses scattering, for a fixed intermediate-state detuning and 1,013-nm Rabi frequency. The scattering from the intermediate state was incorporated through a non-Hermitian Hamiltonian proportional to the scattering rate ($-i\gamma_e|e\rangle\langle e|$). Finally, we inferred an analytical form of the phase and amplitude profiles, which are given by

$$\Omega_{420}(t)/\Omega_{1013} = \Omega_0 + \Omega_1\,\mathrm{sech}[\omega_\Omega\tau]^\alpha,$$
$$\phi(t) = \delta_0\tau + B\tanh(\lambda\tau),$$

in which $\tau = t - T/2$; in principle, one can also add a relative phase offset similar to $\varphi_0$ in equation (1) for further fine-tuning. This ansatz realizes an exact CZ gate for the gate parameters

$$\Omega_0 = 32.7403, \qquad B = 2\pi \times 0.2503,$$
$$\Omega_1 = -31.1404, \qquad \lambda = 0.9372\Omega,$$
$$\omega_\Omega = 0.2668\Omega, \qquad \delta_0 = -0.9491\Omega,$$
$$\alpha = -0.1131,$$

which has a duration of $(\Omega T/2\pi) = 1.207$. We set the reference point such that, for $\Omega_{420} = \Omega_{1013}$, the system is at two-photon resonance and has a two-photon Rabi frequency $\Omega$. This smooth-amplitude gate has advantages of stronger intermediate-state scattering suppression and reduced off-resonant coupling to other states. On the other hand, owing to a larger peak Rabi frequency, this gate is more susceptible to finite-blockade effects. The error budget for both gates can be found in Extended Data Table 1.

Extending beyond two-qubit gates, we find that a slightly more general ansatz allows us to implement a nearly time-optimal three-qubit CCZ gate with the phase profile given by

$$\phi(t) = A_1\sin(\omega_1\tau) + A_2\sin(\omega_2\tau) + B\tanh(\lambda\tau) + \delta_0\tau,$$

in which the Rabi frequency is kept constant and the parameters are

$$A_1 = 2\pi \times 2.1460, \qquad \omega_1 = 0.2101\Omega,$$
$$A_2 = 2\pi \times -0.0719, \qquad \omega_2 = 1.8957\Omega,$$
$$B = 2\pi \times -0.6432, \qquad \lambda = 0.6941\Omega,$$
$$\delta_0 = -1.3646\Omega,$$

which results in an exact gate with duration $(\Omega T/2\pi) = 1.751$. Finally, we note that the same methods can be directly extended to the design of CPHASE($\theta$) and CCPHASE($\theta$) gates, which is important in the context of digital quantum simulation[15]. In Extended Data Fig. 4b,c, we show how the two-qubit gate duration scales with the phase $\theta$.

## Dark states in two-photon Rydberg gates

In this section, we describe the physics associated with the three-level system present in the two-photon transition to the Rydberg state and derive how the Rydberg population can be realized through either the dark or the bright states. The basic intuition can be developed at the single-particle level, at which the system is described by the three-level Hamiltonian in the $\{|1\rangle, |e\rangle, |r\rangle\}$ basis,

$$H = \begin{pmatrix} 0 & \dfrac{\Omega_b}{2} & 0 \\[2mm] \dfrac{\Omega_b}{2} & -\Delta & \dfrac{\Omega_r}{2} \\[2mm] 0 & \dfrac{\Omega_r}{2} & -\delta \end{pmatrix}, \qquad (2)$$

in which we use symbols $\Omega_b := \Omega_{420}$ and $\Omega_r := \Omega_{1013}$ in this section for clarity of expressions. We also assume that the amplitude and phase of the red 1,013-nm laser are kept constant at all times and the blue 420-nm phase is captured by the time-dependent two-photon detuning $\delta := \delta(t) \propto -\phi'(t)$.

At large intermediate detunings ($\Delta \gg \delta, \Omega_{b/r}$), this system is conveniently described in the dark-state basis (as summarized in Extended Data Fig. 2a,b) formed by the eigenstates of equation (2) at the two-photon resonance ($\delta = 0$), which to the leading order in $\Omega_r/\Delta$ is,

$$|D\rangle = -\frac{1}{\sqrt{1+\alpha^2}}\,|1\rangle + \frac{\alpha}{\sqrt{1+\alpha^2}}\,|r\rangle,$$

$$|B\rangle = \frac{\alpha}{\sqrt{1+\alpha^2}}\,|1\rangle + \frac{\sqrt{1+\alpha^2}\,\Omega_r}{2\Delta}\,|e\rangle + \frac{1}{\sqrt{1+\alpha^2}}\,|r\rangle,$$

$$|E\rangle = -\frac{\alpha\Omega_r}{2\Delta}\,|1\rangle + |e\rangle - \frac{\Omega_r}{2\Delta}\,|r\rangle,$$

in which $\alpha = \Omega_b/\Omega_r$. Note that the 'dark state' $|D\rangle$ has no contribution from the intermediate state, the 'bright state' $|B\rangle$ populates the intermediate state $\propto 1/\Delta^2$ and $|E\rangle$ is composed essentially entirely from $|e\rangle$.

For our purposes, the initial state is always $|1\rangle$, which is subsequently dressed by the blue light to $|\tilde{1}\rangle$. This is because the amplitude rise time of the blue laser to its initial value of $\Omega_b(0)$ is on the timescale of 10 ns, which is much slower than the adiabaticity limit set by $\Delta$ and much faster than the two-photon Rabi frequency relevant for populating the Rydberg state; thus, the initial state indeed corresponds to

$$|\tilde{1}\rangle = |1\rangle + \frac{\alpha \Omega_r}{2\Delta} |e\rangle$$
$$= \frac{\alpha}{\sqrt{1+\alpha^2}} |B\rangle - \frac{1}{\sqrt{1+\alpha^2}} |D\rangle + O(\Delta^{-3})|E\rangle,$$

which is well supported on the $\{|D\rangle, |B\rangle\}$ states alone. Moreover, the excited state $|E\rangle$ is detuned from the other two by $\Delta$ and all direct couplings to it are on the order of $\Omega_r/\Delta$; thus any population transfer out of the $\{|D\rangle, |B\rangle\}$ manifold will be suppressed by $(\Omega_r\delta)^2/\Delta^4$ and the subsequent evolution of state $|\tilde{1}\rangle$ is described by an effective two-level system (Extended Data Fig. 2b). In this picture, the energy splitting is set by the AC Stark shift (diagonal terms) and the effective off-diagonal coupling is given by a combination of the two-photon detuning and diabatic terms (off-diagonal terms). Crucially, the Rydberg-state population can be realized in many inequivalent ways; for example, for $\alpha = 1$, states of the form $\sqrt{1-\beta} |B\rangle + \sqrt{\beta} |D\rangle$ have the same Rydberg population for $\beta$ and $\beta \to 1 - \beta$ (in which $\beta \in [0, 1]$), despite very different intermediate-state contributions.

First, consider the case of the parameterized time-optimal gate in which the blue Rabi frequency is kept constant throughout the duration of the gate ($\alpha(t) = 1$), up to the finite rise and fall times. The initial state is simply $(|D\rangle - |B\rangle)/\sqrt{2}$ and the Hamiltonian is equivalent to

$$\widetilde{H}_{\text{to}} = \begin{pmatrix} 0 & -\delta/2 \\ -\delta/2 & \frac{\Omega_r^2}{2\Delta} \end{pmatrix},$$

in which the magnetic field sign is decided by $\Delta$ and the phase of the Rabi frequency is given by the sign of the two-photon detuning $\delta$. The time evolution under this Hamiltonian (which corresponds to driving the two-photon transition) can be solved exactly for fixed $\delta$ and the population in the dark state is

$$P_{\text{D}} = \frac{1}{2} - \delta\Delta \frac{\Omega_r^2 \sin^2\left(\frac{\sqrt{4\delta^2\Delta^2 + \Omega_r^4}}{4\Delta}t\right)}{2\delta^2\Delta^2 + \Omega_r^4},$$

which can go above or below 1/2, depending on the relative sign of the detunings. More precisely, the Rydberg population is realized predominantly by means of the dark state when $\delta\Delta < 0$ ($P_{\text{D}} > 1/2$), that is, when the intermediate-state detuning $\Delta$ and the two-photon detuning $\delta$ have opposite signs; for a time-dependent detuning, the relevant sign is the one at the beginning of the pulse. In Extended Data Fig. 2c, we plot the intermediate-state population and the Bloch-sphere trajectories for the time-optimal gate at two different signs of the two-photon detuning $\delta$. As expected, one of the trajectories realizes the Rydberg population through the dark state and, as a result, minimizes the intermediate-state population, leading to suppressed scattering.

The remaining scattering comes mostly from the large admixture of the bright state in the initial state and can be further reduced by using a smooth-amplitude profile, which starts at low blue Rabi frequency ($\alpha \ll 1$) and only later increases to larger values, as is the case in gate schemes based on the adiabatic passage[29]. Note that operating at a fixed lower $\alpha$ does not further reduce scattering because a larger admixture of the bright state is necessary to realize the same integrated

Rydberg population as before. In Extended Data Fig. 2d, we show the intermediate-state population and the effective Bloch-sphere trajectory for the smooth-amplitude gate introduced in the previous section. This gate occupies the instantaneous dark state for most of its execution time, admixing only as much bright state as is necessary to realize the required Rydberg population. We find numerically that the degree of scattering suppression depends on the speed relative to the time-optimal gate (assuming fixed 1,013-nm Rabi frequency): scattering is suppressed by a factor of 1.2 (relative to the time-optimal gate) if the two gates take the same time and by a factor of roughly 2.5 if the smooth-amplitude gate is twice as long as the time-optimal gate. This tunability is useful for choosing a gate based on the dominant error source: a slower gate could be more beneficial when scattering dominates, whereas a faster gate can be used when $T_2^*$ or Rydberg decay is the main source of errors.

We note that, despite the presence of several intermediate states (Extended Data Figs. 1 and 3a), which are at slightly different detunings and couple with different Rabi frequencies, the dark-state picture remains valid. We find that this is the case numerically and note that the intermediate-state population plots in Extended Data Fig. 2c,d include contributions from all intermediate states in a numerical model realistic for $^{87}$Rb.

**Simulating two-qubit gate error sources**

The level diagram in Extended Data Fig. 3a summarizes the atomic-level structure used for numerical modelling, as well as the assumed scattering rates, lifetimes, branching ratios, Rabi frequencies and detunings. We model scattering and Rydberg lifetime by performing a full density-matrix simulation of the two atoms with the eight levels depicted in Extended Data Fig. 3a (including three intermediate states). Our modelling also explicitly includes the coupling to the other ($m_J = -1/2$) Rydberg state, which is 24 MHz lower in energy and is driven with a Rabi frequency suppressed by a factor of three (owing to Clebsch–Gordan coefficients). For the gate, we assume approximately 20 ns min–max rise/fall times of our AOM pulse profile (in a Blackman profile), which has substantial implications for the off-resonant coupling to the adjacent Rydberg state, in terms of whether it is adiabatically 'dressed' or diabatically occupied. Because the impact of the other Rydberg state on the gate fidelity depends on the details of the pulse power profile and degree of calibration, we report a range of values that is reasonable for the assumptions mentioned above.

Finite temperature is also assumed in our error modelling, in which—for our given temperature—we calculate the position spread of the atom in a trap and the corresponding fluctuation in interaction strength from the distance-dependent blockade interaction $V_{\text{Ryd}} \propto r^{-6}$. We note that finite temperature can also contribute to single-qubit dephasing through both velocity spread and photon recoil[92], but these effects are already encompassed by our single-qubit ground-Rydberg $T_2^*$ measurements. The $T_2^*$ can also have contributions from other phenomena, such as electric-field fluctuations and fluctuations in the 1,013-nm light shift (which has a differential light shift of about 20 MHz on the ground-Rydberg transition), and so, for our simulations, we simply use the measured $T_2^* = 3$ μs assuming a Gaussian distribution of detunings.

**Projecting path to 99.9% and error breakdown**

We can also use our detailed microscopic error modelling, which reproduces similar fidelities as the measured 99.5%, to project future performance. To reach 99.9% fidelity, the sum of the errors in Extended Data Table 1 needs to be suppressed to below 0.1%, which can be achieved by, for example, going two times further detuned, having a three times longer Rydberg lifetime (for instance, exciting to a higher $n$ state), two times longer $T_2^*$ (note that dephasing error scales as $\propto 1/(\Omega T_2^*)^2$) and suppressing coupling to the other $m_J$ state. This suppression can be achieved by applying a larger magnetic field, using the smooth-amplitude gate or eliminating coupling altogether by exciting

from a stretched state or through the $6P_{1/2}$ excited state. An alternative approach to reaching 99.9% fidelity could be going to three times higher Rabi frequency (again while suppressing coupling to the other $m_J$ state) and two times larger detuning. Other unique opportunities towards future improvements include single-photon excitation to the Rydberg state, which circumvents intermediate-state scattering but has higher Doppler and recoil sensitivity[79,82,92], and has been explored in a variety of contexts with both alkali[93,94] and alkaline-earth(-like) atoms[46,47,51].

Separately, this microscopic error analysis can also be used to analyse the type of error produced, that is, whether it is a Pauli (X, Y, Z) error, atom loss or leakage to other $m_F$ states. Such an understanding is particularly important for quantum error correction[36,45], for which neutral atoms have various unique opportunities[95,96], as knowing the noise structure can be used to enhance the performance of error-correcting schemes. Our present modelling suggests that most errors are Z-type and loss/leakage-type errors, as previously highlighted in ref. 36. If atom loss is directly detected, these errors would constitute a so-called erasure error[45] and, moreover, atom loss in this case is in fact a biased erasure error because almost all of it originates from state $|1\rangle$, as pointed out and developed in ref. 96. Alkaline-earth(-like) atoms are particularly well suited to erasure conversion, owing to their metastable qubit structure[45–47]. In Extended Data Table 1, we summarize how each error source breaks down into the five error types mentioned above and find that only the scattering and Rydberg decay errors can lead to X and Y Pauli errors. For simplicity, we estimate the effective single-particle error channel; that is, we compute the process matrix for the two-qubit gate and then trace out one of the qubits. The full process matrix can be used to study more complicated properties of this Pauli + loss/leakage noise model, such as correlations.

## Gate calibration and benchmarking

We calibrate the gate using the global randomized benchmarking method shown in Fig. 3. In Extended Data Fig. 5, we show an example sequential optimization of CZ gate parameters for the time-optimal gate and the smooth-amplitude gate, leading up to the measurements in Fig. 3b. Once found, these gate parameters are empirically optimal for all the other benchmarking methods, such as the Bell-state measurements in Fig. 2, and are consistent from day to day.

The qubits also pick up a global single-particle phase during the gate, which we cancel here using global $X$ gates between pairs of CZ gates for simplicity. For many applications, such as quantum error correction, our gates are naturally used in this configuration (as was done for the quantum circuits implemented in ref. 4). We also further calibrate and benchmark the CZ gate without the $X$ gate, by performing randomized benchmarking composed of repeated application of CZ gates and random single-qubit rotations, as shown in Extended Data Fig. 6a. Here the final several CZ gates and random rotations are calculated to return the qubit pair from the resulting entangled state back to the initial product state. We perform a single-qubit $Z$ rotation after each CZ gate to compensate for the accumulated single-particle phase, which we calibrate as simply another parameter of the gate to scan and optimize. As well as calibrating this single-qubit $Z$ rotation, this circuit also benchmarks a gate fidelity of 99.48(2)% on 20 atoms. To measure the raw Bell-state data in Fig. 2b, we used both the CZ gate calibration by means of the first method of global randomized benchmarking and this single-particle phase calibration, as two independent calibration stages.

We note that all of our randomized benchmarking methods use only global rotations for simplicity. The symmetry introduced by global rotations makes us less sensitive to certain types of error[97], SWAP being an extreme example, which we expect to be negligible. Nonetheless, because the atoms are identical and placed very close together, there is a large degree of symmetry between the two qubits and we expect this global benchmarking procedure to faithfully capture our gate fidelity, which we confirm with numerical simulations. This is further confirmed by the fact that the experimentally extracted fidelity is consistent between this method and the Bell-state measurement. Quantitatively, in Extended Data Fig. 3e, we simulate all the benchmarking methods, including the full randomized benchmarking protocol, using the microscopic error model developed in this work. We find that all methods give consistent results, with the Bell-state fidelity lower-bounding the other curves.

Here we describe some experimental procedures used while taking data. First, for the benchmarking curves involving varying numbers of CZ gates, we take data in a cyclic manner to avoid systematic biases that could be introduced by experimental drift (for example, alternating in the sequence of 20, 0, 16, 4, 12 and 8 CZ gates for the data taken in Fig. 3). We perform several rounds of this cyclic sequence in one continuous stretch of time (over roughly a few hours for each plot in Figs. 2 and 3). For each gate number, we average over 300 sets of random rotations.

To extract a gate fidelity, we fit our data to exponential decays. We note that, as we are mostly in the linear regime of the exponential curve, adding an offset to the fit (and then rescaling the fitted exponent, as done in some randomized benchmarking works) has a negligible effect on the extracted fidelity and so we fit to an exponential decay without an offset.

## Bell-state fidelity

Here we outline the method used for the Bell-state data in this work. We measure the Bell-state fidelity as the average of coherences and populations[5]. The coherence is extracted by measuring the amplitude of parity oscillations, using the circuit in Fig. 2c. The populations are calculated as the sum of the $|00\rangle$ and $|11\rangle$ states, which we correct for further atom loss as described below. The Bell-state populations can be overestimated owing to atom loss contributing to the perceived detection of state $|11\rangle$ (ref. 5), because loss shows up identically as $|1\rangle$ in our state-detection procedure. To account for this, here we measure the atom loss probability when applying the sequence of gates (by turning off the pushout of state $|1\rangle$ for state discrimination), to find the extra contribution of atom loss to the Bell-state population. To perform this loss subtraction, we subtract the observed $|11\rangle$ (that is, observed loss of both atoms) population during the loss measurement directly from the measured populations, as well as measuring the loss-per-atom-per-gate, which can also contribute to state $|11\rangle$ by converting $|01\rangle$ and $|10\rangle$ to $|11\rangle$. This loss subtraction is performed for Fig. 2d. We emphasize that this correction strictly lowers the measured gate fidelity (without applying this loss subtraction, the measured CZ gate fidelity on the raw Bell-state fidelity data is extracted to be 99.57(4)%).

We next evaluate a SPAM-corrected Bell-state fidelity from the measured raw Bell-state fidelity of 98.0(2)% after a single gate in Fig. 2b. To extract a SPAM-corrected Bell-state fidelity, we first measure relevant SPAM errors. In particular, we measure a population of 99.6(1)% in state $|0\rangle$ when we try to prepare into $|0\rangle$ and, likewise, a population of 99.4(1)% in state $|1\rangle$ after state preparation into $|1\rangle$. These measurements include further effects from loss and imaging/pushout fidelity. Specifically, there is a 0.35(5)% probability that an atom is lost during the sequence and the gate itself causes 0.17% further loss on top of this baseline loss. Our pushout fidelity of 99.83(1)% affects the measurement fidelity of $|1\rangle$, for which we also correct. From these measurements, we estimate the amount of population leaked into other $m_F$ levels during state preparation, as well as the probability of atom loss both before and after the circuit. From these values, we follow the method described in ref. 5 to extract a SPAM-corrected Bell-state fidelity of 99.4(4)%.

## Analysis of correlations between gate sites

Here we further analyse our data to characterize whether gate errors across the array are correlated. We study the 20-atom and 60-atom global randomized benchmarking data from Fig. 3 and consider the distribution of the number of errors that occurs in each experimental

shot, in which an error is defined as whenever a qubit pair does not return to the initial $|00\rangle$ product state. In Extended Data Fig. 7a,c, we plot the number of errors occurring in each shot as a function of the number of CZ gates applied, in which the mean number of errors grows owing to the 0.5% error per CZ gate. We compare our data (bottom) with a model consisting of a Poissonian distribution of errors centred at the experimental mean (top). We find that the Poisson distribution model approximates our data and large-scale correlated errors are not common in our system.

To analyse more quantitatively in a single plot, we average the data for all numbers of gates and plot the resulting distribution in Extended Data Fig. 7b,d. We find that higher weight errors for both the 20 atoms and 60 atoms data are greatly suppressed. More quantitatively, we compare our data to the average of the Poissonian distributions plotted in Extended Data Fig. 7a,c and find small deviation from the Poisson distribution. We find that these data are better described by a model in which the CZ gate fidelity is sampled from a Gaussian distribution in each shot (which would arise from, for example, global shot-to-shot fluctuations in detuning, captured by our $T_2^*$).

In Extended Data Fig. 8a,c, we plot the covariance matrix between gate sites for the return to the initial state $P_{|00\rangle}$, after 0 CZ gates and after 20 CZ gates, qualitatively observing evidence of small positive covariance between nearby gate sites. In particular, Extended Data Fig. 8b,d shows the growth of covariance between neighbouring gate sites as a function of the number of CZ gates applied. These observations are consistent with known physical effects related to our error budget, namely, Rydberg lifetime and $T_2^*$. For example, decay of Rydberg atoms to nearby $P$ states during the gate can cause detuning shifts owing to strong long-range interactions between $S$ and P Rydberg states (decaying as $1/R^3$), as well as hopping of the $P$ state to nearby gate sites. Furthermore, fluctuations in Rydberg 1,013-nm beam intensity can give rise to shot-to-shot fluctuations in gate fidelity with a spatial dependence. Classical Monte Carlo simulations of these two phenomena (not plotted here) reveal that both error sources can result in non-zero covariance, which seems to be described by quadratic growth for a small number of gates but linear asymptotically. We note that the crosstalk between gate sites, on the scale of 10 kHz, should be negligible.

This measurement of covariance after 20 gates is a highly sensitive probe to small correlations between gate sites building up over the course of the circuit. After one applied gate, this covariance seems to be smaller than other main error sources, so these correlations will have little effect for quantum circuits in which atoms are involved in gates not just at a single gate site but across the entire entangling zone. Moreover, when running quantum circuits using atom transport[4], the approximately 100-μs delay between subsequent gates would result in leftover Rydberg states being expelled from the array, completely suppressing the effect from Rydberg decay described above.

## CCZ gate design
To find time-optimal gates for the multi-qubit controlled phase gates, such as the CCZ gate, we use the optimal control methods similar to ref. 8. The gates in ref. 8 are found by looking for two-qubit diagonal gates up to global single-qubit $Z$ rotations. However, for more than two qubits, there are several distinct ways of realizing the controlled-$Z$ gates that are not connected by a $Z$ rotation but rather by a general single-qubit rotation, and these various gate realizations can be different. We use the approach in which the controlled phase flip is applied to the $|000\rangle$ state to find multi-qubit controlled $Z$ gates for a larger number of qubits; we present the obtained times in Fig. 4e and Extended Data Fig. 9a. In Extended Data Fig. 9b, we show the time-optimal pulse profiles of multi-qubit CZ gates up to the six-qubit CCCCCZ. Finally, we note that an analytical ansatz (defined in the section dedicated to parameterized gates) similar to that used for the CZ gate allows us to parameterize the three-qubit controlled-$Z$ gate with only a marginal

decrease in speed ($\Omega T/2\pi = 1.75$). In fact, we find that this ansatz can also realize the CCCZ gate and we expect simple generalizations to be capable of realizing these gates for even larger numbers of qubits.

## GHZ states
To characterize the CCZ gate experimentally, we create a GHZ state using the circuit shown in Extended Data Fig. 9c. In Extended Data Fig. 9d, we generate GHZ states after application of two CCZ gates, with populations of 92.9(3)% and parity contrast of 89(1)%, giving a raw GHZ-state fidelity of 90.9(6)% (without any loss subtraction). For Fig. 4, we calibrate the gate by repeatedly applying the CCZ-π-CCZ part of the circuit such that after six and ten CCZ gates, we generate GHZ states with reduced fidelity and we observe a 2.1(2)% reduction in the raw GHZ fidelity as a function of the number of CCZ gates applied. For the data in Fig. 4d, we operate at 7.8-GHz intermediate-state detuning and 3.9-MHz Rabi frequency.

## Data availability
The data that support the findings of this study are available from the corresponding author on reasonable request.

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

**Acknowledgements** We thank S. Hollerith for discussions on error contributions from motional states and M. Cain for insights on atomic dark states. We further thank M. Abobeih, H. Bernien, N.-C. Chiu, S. Geier, J. Guo, A. Keesling, H. Pichler and P. Stroganov for useful discussions, technical support and careful reading of the manuscript. We also thank QuEra Computing and IPG Photonics and in particular N. Gemelke, M.-G. Hu, M. Kwon and A. Lukin for support in the development and testing of the high-power 1,013-nm Rydberg laser. We acknowledge financial support from the U.S. Department of Energy (DOE Quantum Systems Accelerator Center, contract numbers 7568717 and DE-SC0021013), the Center for Ultracold Atoms, the National Science Foundation, the Army Research Office MURI (grant number W911NF-20-1-0082) and the DARPA ONISQ programme (grant number W911NF2010021). S.J.E. acknowledges support from the National Defense Science and Engineering Graduate (NDSEG) fellowship. D.B. acknowledges support from the NSF Graduate Research Fellowship Program (grant DGE1745303) and the Fannie and John Hertz Foundation. T.M. acknowledges support from the Harvard Quantum Initiative Postdoctoral Fellowship in Science and Engineering. N.M. acknowledges support by the Department of Energy Computational Science Graduate Fellowship under award number DE-SC0021110.

**Author contributions** S.J.E., D.B., M.K., S.E., T.M., H.Z., S.H.L. and A.A.G. contributed to the building of the experimental setup, performed the measurements and analysed the data. M.K. developed the two-qubit gate schemes and performed theoretical analysis. N.M. and M.K. developed the multi-qubit gate schemes. T.T.W., H.L. and G.S. contributed to initial developments and insights into gate error sources. All work was supervised by M.G., V.V. and M.D.L. All authors discussed the results and contributed to the manuscript.

**Competing interests** M.G., V.V. and M.D.L. are co-founders and shareholders and H.Z. is an employee of QuEra Computing.

**Additional information**
**Correspondence and requests for materials** should be addressed to Mikhail D. Lukin.

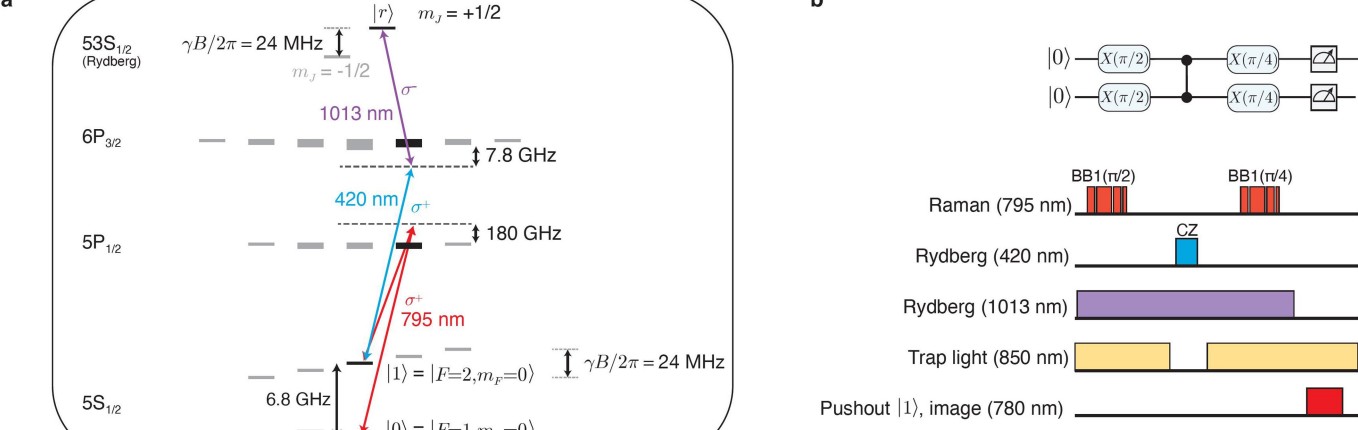

**Extended Data Fig. 1 | Atomic-level diagram and pulse sequence. a**, Level diagram showing key levels of $^{87}$Rb used in our quantum circuits. The clock states, $|0\rangle$ and $|1\rangle$, are the qubit states used in this work. Excitation to the Rydberg state between $|1\rangle$ and $|r\rangle$ is carried out by a two-photon transition driven by 420-nm and 1,013-nm lasers. Single-qubit rotations are realized with an amplitude-modulated 795-nm laser that drives Raman rotations between the $m_F = 0$ clock states. A DC magnetic field of 8.5 G is applied throughout this work. The Rydberg detuning signs and polarization signs are carefully selected for various optimizations: for suppressing 420-induced differential light shift between $|0\rangle$ and $|1\rangle$, we red-detuned the $6P_{3/2}$ transition; for using dark-state physics (nominally the phase profile corresponds to a certain sign of two-photon detuning), we thereby choose positive two-photon detuning, which—in turn— then suppresses coupling to $m_J = -1/2$ by being primarily on the upper side of $m_J = +1/2$; and, finally, the 1,013-nm light shift is lower (by about 30%) at this single-photon detuning sign, as there is a magic wavelength of about 1 GHz red-detuned of $6P_{3/2}$ for the $|1\rangle \rightarrow 53S_{1/2}$ transition[98]. Two downsides of this detuning choice are that this choice of 420-nm polarization and detuning causes a vector light shift in the hyperfine ground-state manifold that causes the $m_F$ levels to be pushed closer together, as opposed to further apart, which could exacerbate effects arising from 420-induced vector light shifts coupling adjacent $m_F$ states in the ground-state manifold (although negligible here), and the other downside is that the $m_J = -1/2$ Rydberg pair states are closer detuned to the two-photon excitation and so we require a larger interaction strength to suppress their excitation, although the matrix element to these states is smaller. **b**, Example pulse sequence, here for making a $|\Phi^+\rangle$ Bell state between two qubits. Traps are pulsed off for a few hundred ns during the Rydberg gate to avoid both antitrapping of the Rydberg state and inhomogeneous light shifts that broaden the transition, and the ground-state atoms are then recaptured for roughly 3 μs between consecutive gate applications.

**a**

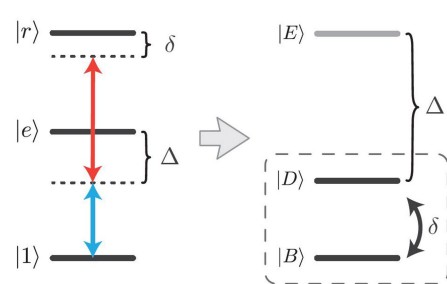

**b**

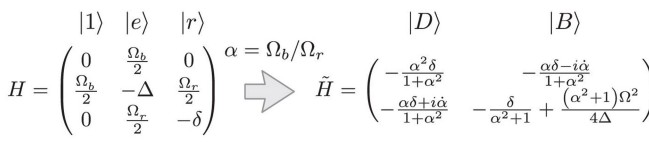

$$H = \begin{pmatrix} 0 & \frac{\Omega_b}{2} & 0 \\ \frac{\Omega_b}{2} & -\Delta & \frac{\Omega_r}{2} \\ 0 & \frac{\Omega_r}{2} & -\delta \end{pmatrix} \quad \alpha = \Omega_b/\Omega_r \quad \tilde{H} = \begin{pmatrix} -\frac{\alpha^2 \delta}{1+\alpha^2} & -\frac{\alpha\delta - i\dot{\alpha}}{1+\alpha^2} \\ -\frac{\alpha\delta + i\dot{\alpha}}{1+\alpha^2} & -\frac{\delta}{\alpha^2+1} + \frac{(\alpha^2+1)\Omega^2}{4\Delta} \end{pmatrix}$$

**c**

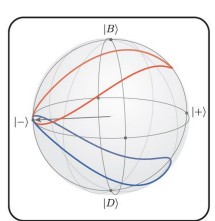

**d**

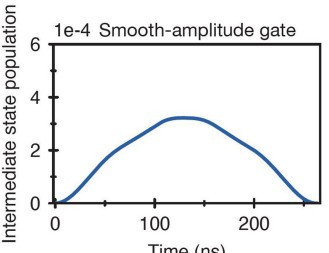

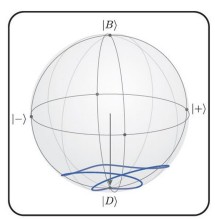

**Extended Data Fig. 2 | Dark-state physics in two-photon Rydberg gates.**
**a**, In the far-detuned limit, the three-level system can be effectively described as a two-level system with a 'dark' state $|D\rangle$ and a 'bright' state $|B\rangle$. The population in the excited state $|E\rangle$ is suppressed by a factor $\propto \Delta^{-4}$ and does not participate in system dynamics. **b**, The Hamiltonian of the bare atomic system and the effective two-level system, in which $\alpha$ is the (time-dependent) ratio between the blue and red Rabi frequencies and $\dot{\alpha}$ is its time derivative. **c**, Intermediate-state population during the parameterized time-optimal gate in the dark and bright configurations together with their Bloch-sphere trajectories in the $\{|D\rangle, |B\rangle\}$ basis. **d**, Intermediate-state population during the smooth-amplitude gate and its Bloch-sphere trajectory in the instantaneous $\{|D\rangle, |B\rangle\}$ basis. The simulation parameters correspond to those mentioned in Extended Data Fig. 3a and Extended Data Table 1.

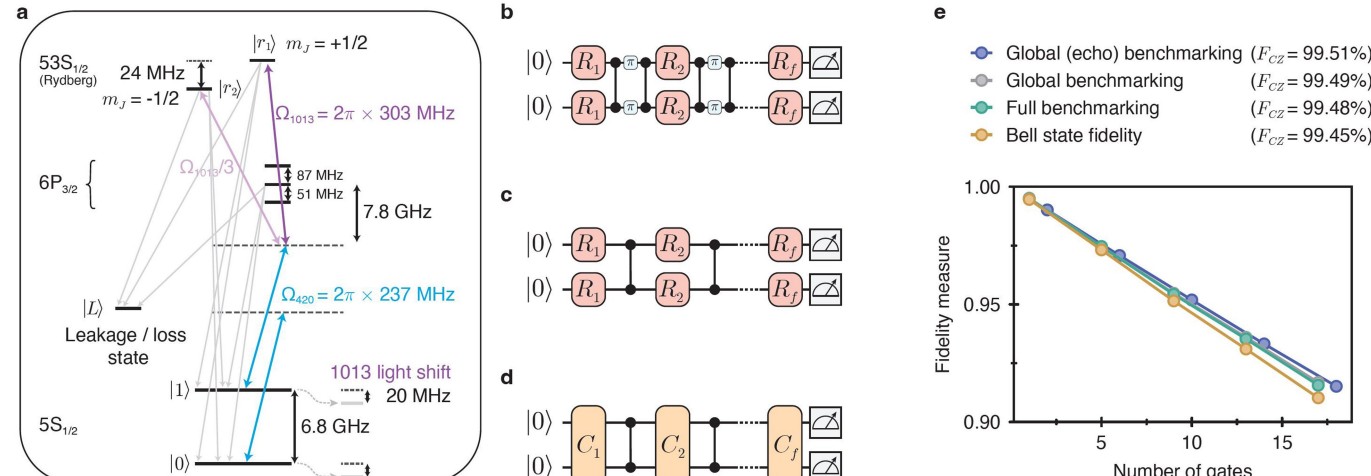

**Extended Data Fig. 3 | Atomic physics error-level diagram and numerical comparison of benchmarking methods. a**, Level diagram shows the eight states assumed in the simulation. We assume a 88-µs Rydberg-state lifetime (based on measured $T_1$ with 1,013-nm scattering lifetime subtracted) and a 110-ns lifetime for the intermediate states. We assume the following branching ratios for the intermediate states[99]: $\eta_{e \to L} = 0.6142$, $\eta_{e \to 1} = 0.2504$, $\eta_{e \to 0} = 0.1354$, and the following ones for the Rydberg states[100]: $\eta_{r \to L} = 0.894$, $\eta_{r \to 1} = 0.053$, $\eta_{r \to 0} = 0.053$. We use the branching ratios between different channels of intermediate-state scattering as reported in ref. 99 and we also assume a simplified model in which all indirect paths (through $4D$ and $6S$) populate the ground-state manifold uniformly. The Rydberg lifetime has both radiative decay (170 µs) and black-body decay (128 µs) components, which we obtain by rescaling the values in ref. 100 to $n = 53$. The microwave component results purely in atom loss and we assume

that the radiative decay populates the ground-state manifold uniformly. We note that a more accurate treatment of the decay channels[36] could increase error modelling precision in future work. **b**, Benchmarking of the CZ-π-CZ sequence with global random rotations, which is insensitive to the single-particle phase. **c**, Benchmarking a standalone CZ gate with global random rotations, which enables separate calibration of the single-particle phase. **d**, The usual interleaved randomized benchmarking method using random two-qubit Clifford gates (not performed in this work). **e**, Numerical simulation of the presented benchmarking methods and the Bell-state-preparation method, using the realistic error model developed in this work. All approaches give consistent results, with the Bell-state fidelity measurement lower-bounding the other curves.

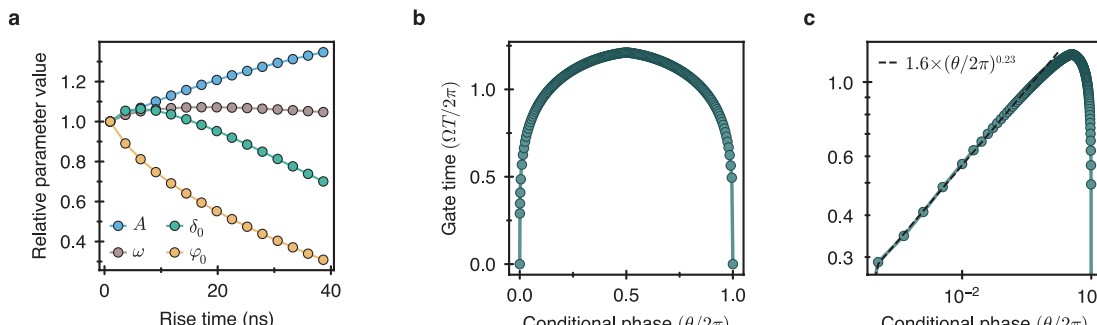

**Extended Data Fig. 4 | Robust and versatile two-qubit gates. a**, Robustness to experimental offsets. For some systematic experimental offsets, such as finite rise time of the 420-nm Rydberg laser pulse, an exact CZ gate can still be found. The relative values of gate parameters for the time-optimal gate are plotted as a function of pulse rise time. For no rise time, the parameter values used here are: $A/2\pi = 0.0988$, $\omega/\Omega = 1.3629$, $\varphi_0 = -2.6082$ and $\delta_0/\Omega = -0.0187$. **b**, Duration of a controlled-phase gate CPHASE($\theta$). The CZ gate ($\theta = \pi$) is the longest in this gate family. Because faster gates are expected to have higher fidelity, an average CPHASE gate should perform even better than the CZ gate reported in this work, which is an exciting perspective for near-term digital simulation. **c**, Plotting on a log–log plot, we see that, for small angles $\theta$, the gate time decreases with an approximate power law $\Omega T(\theta)/2\pi = 1.6 \times (\theta/2\pi)^{0.23}$. This suggests that applying very small phases can be costly and should be taken into account when designing digital simulation schemes. Although these small-angle gates are time-optimal when applying a single, fixed-amplitude pulse, different approaches could perform better. Exploring other exact and approximate gate schemes, such as Rydberg dressing and a detuned 2π pulse, in the small-angle regime is an interesting direction for future work.

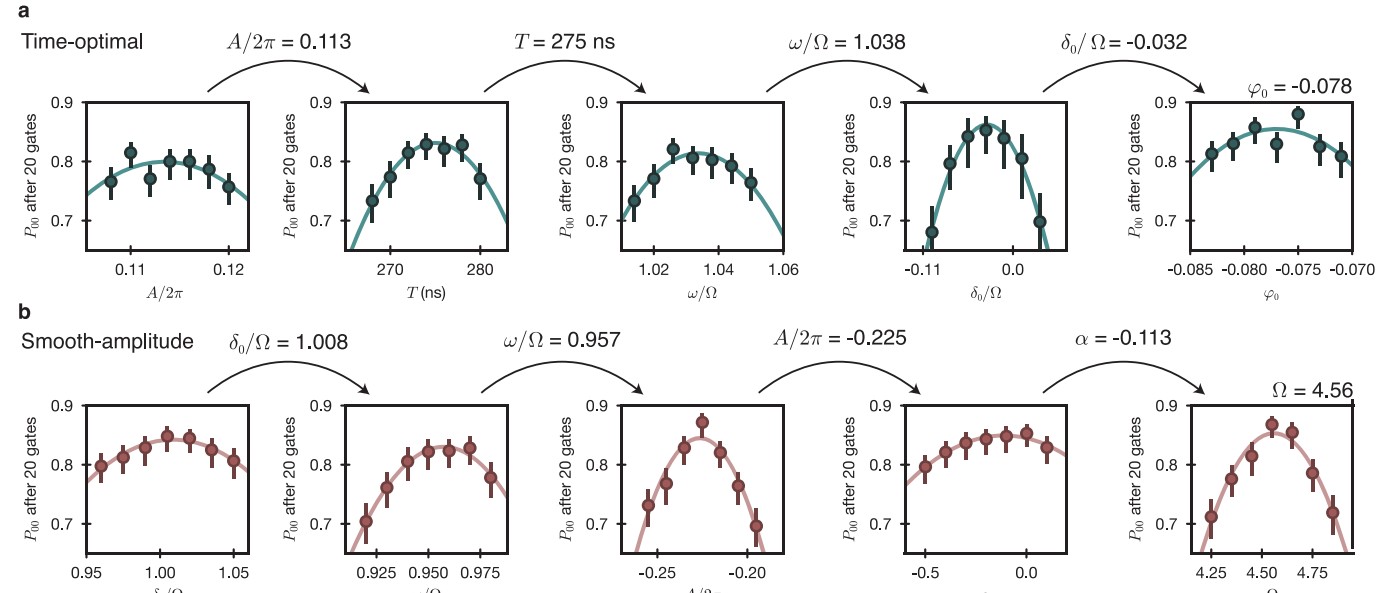

**Extended Data Fig. 5 | Empirical optimization of two-qubit gate fidelity.**
**a**, Calibrating gate parameters for the time-optimal gate, indicating the chronological sequence of calibrations performed before measurement in Fig. 3b. We scan individual gate parameters and measure the probability of return to the initial state after application of 20 entangling gates. **b**, Analogous calibration of gate parameters for the smooth-amplitude gate, which we performed in the sequence shown before the smooth-amplitude measurement in Fig. 3b. Additional calibration of other gate parameters was performed before these measurements.

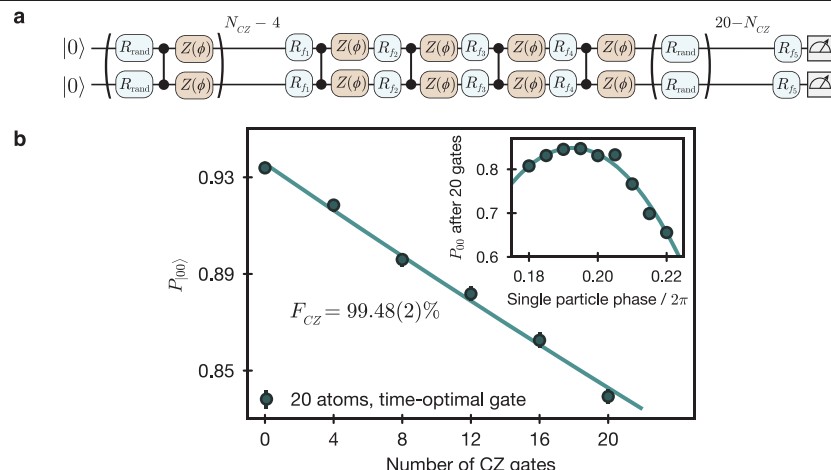

**Extended Data Fig. 6 | CZ gate single-particle phase calibration and benchmarking. a**, Digital circuit for global randomized benchmarking method used to calibrate single-particle phase, in which a $Z$ rotation can be performed after each CZ gate to compensate for the acquired phase. $R_{\text{rand}}$ are single-qubit rotations sampled from a Haar-random distribution and the five rotations $R_{\text{f}}$ are computed to return the atom pair to the initial product state in the absence of gate errors. For the 0 CZ gates point, 20 random rotations are applied, as well as a final rotation precomputed to return population to the initial state. **b**, Experimental data used for calibrating the single-particle phase by optimizing the return probability $P_{|00\rangle}$ after 20 CZ gates (inset). For the optimal choice of $\phi$, we extract a 99.48(2)% CZ gate fidelity, fitting to an exponential decay.

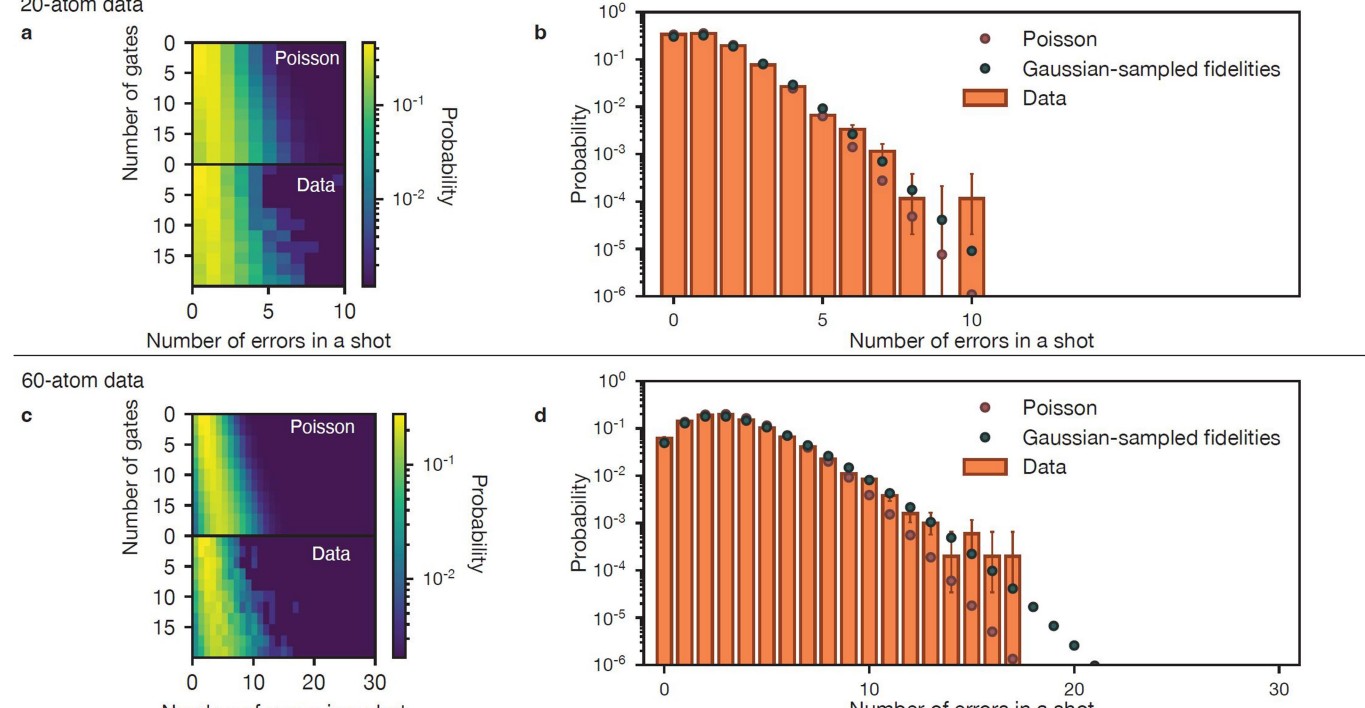

**Extended Data Fig. 7 | Correlated errors in experimental shots. a**, Distribution of errors in each experimental shot as a function of the number of CZ gates applied for the 20-atom data in Fig. 3, showing qualitative agreement with a Poisson distribution centred at the experimental mean for the number of errors in a shot. **b**, Histogram of the number of errors in a shot, averaging over all numbers of gates for the 20-atom data. We compare to one model assuming a Poissonian distribution of errors about the mean, finding some deviation from our data. In a second model, we consider that, in each shot, there is a slightly different gate fidelity, sampled from a Gaussian distribution with a mean of 99.54% and standard deviation of 0.3%. This second model seems to better capture our data. **c**, Repeating the analysis for the 60-atom data in Fig. 3, we notably find no shot (out of the 5,053 total repetitions) with 18 or more errors out of the 30 gate sites. Again, the data are qualitatively similar to a Poisson distribution model. **d**, Averaging over the 60-atom data for all numbers of gates, we find again a small quantitative deviation between the data and a model with a Poisson distribution of errors in each shot. The data seem to be better described by a model in which, in each shot, we sample fidelities from a Gaussian distribution with a mean of 99.5% and a standard deviation of 0.3%.

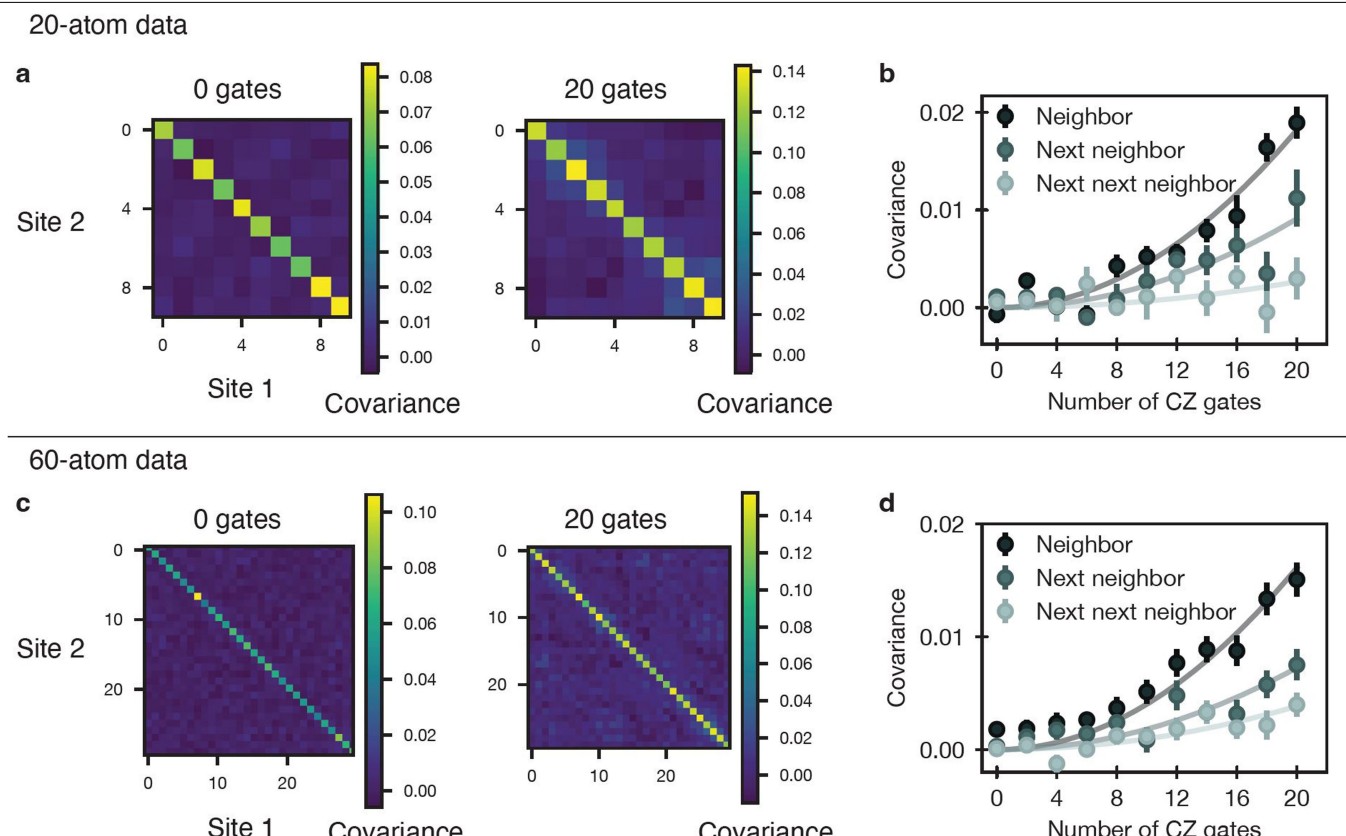

20-atom data

**a**

0 gates

20 gates

**b**

60-atom data

**c**

0 gates

20 gates

**d**

**Extended Data Fig. 8 | Correlations between gate sites. a**, Covariance matrices for the 20-atom data in Fig. 3b after 0 gates and 20 gates, in which local correlations appear after 20 gates. **b**, Covariance averaged over all neighbours, next nearest neighbours and next next nearest neighbours, as a function of the number of CZ gates applied. As a guide to the eye, data are fit to quadratic curves. **c**, Covariance matrices for 60-atom data in Fig. 3d for 0 gates and 20 gates. **d**, Plotting covariance for neighbouring sites for the 60-atom data. Once again, the covariance between nearby sites exhibits small growth throughout the 20 CZ gates applied, in particular for the nearest-neighbour sites.

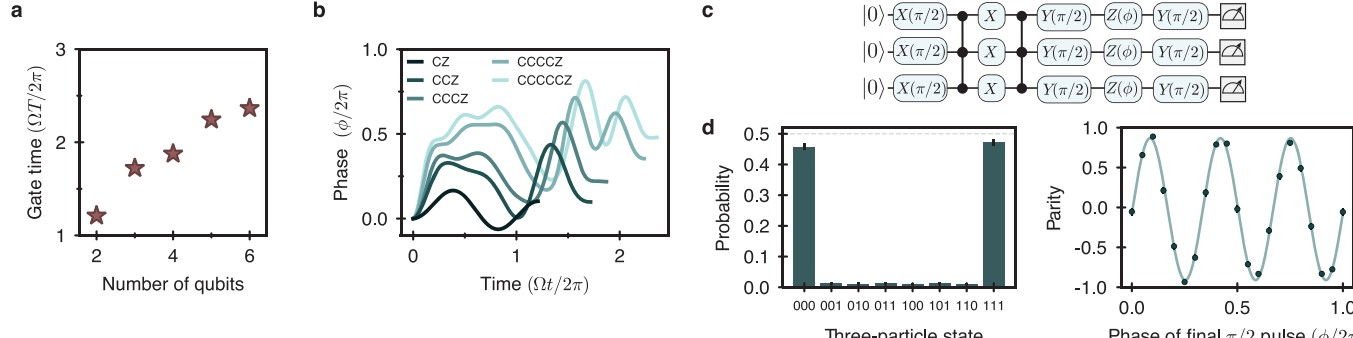

**Extended Data Fig. 9 | Time-optimal pulses for multi-qubit controlled phase gates and GHZ-state data. a**, The execution time of a $C_N Z$ blockade gate as a function of the number of qubits. The $N$-qubit gates are realized by applying a phase flip to the $|0\rangle^{\otimes N}$ state, which is not equivalent to the method of ref. 8 for $N > 2$. For the CCZ gate, applying a $\pi$ phase to the $|111\rangle$ state ($|111\rangle \rightarrow -|111\rangle$), while leaving all other basis states invariant, is related by a global bit-flip to applying a relative $\pi$ phase to the $|000\rangle$ state; however, the two implementations are not equivalent up to a global $Z$ rotation, contrary to the two-qubit case. The time-optimal CCZ gate using the second approach realizes the CCZ gate about 34% faster with ($\Omega T/2\pi$) = 1.72, as compared with ($\Omega T/2\pi$) = 2.61 from the first approach. The two approaches are different because the relative phase of $\pi$ is accumulated between different basis states, which have different rates of phase accumulation. In the case of applying $|111\rangle \rightarrow -|111\rangle$, the

states with the slowest relative rate are $|111\rangle$ and $|011\rangle$, which are driven with the Rabi frequencies of $\sqrt{3}\,\Omega$ and $\sqrt{2}\,\Omega$, respectively, resulting in the phase accumulation rate proportional to $(\sqrt{3} - \sqrt{2})\Omega \approx 0.32\Omega$. By contrast, when the relative phase is applied on the state $|000\rangle$, the smallest accumulation rate is given by the $|001\rangle$ state, which is driven with the Rabi frequency $\Omega$. In general, an arbitrary global single-qubit rotation at the end of the gate can be included to incorporate all of the above approaches in the optimization procedure. **b**, Time-optimal phase profiles (without analytic parameterization) for the $C_N Z$ gates up to six qubits realized by applying a phase flip to the $|0\rangle^{\otimes N}$ state. **c**, Circuit used to generate the GHZ state $(|000\rangle + |111\rangle)/\sqrt{(2)}$ after two CCZ gates. **d**, GHZ states measured experimentally on applying this circuit to seven three-qubit groups in parallel, with populations in $|000\rangle$ and $|111\rangle$ of 92.9(3)% and a parity contrast of 89(1)%, giving a raw GHZ-state fidelity of 90.9(6)%.

**Extended Data Table 1 | Simulated error budget for two-qubit CZ gates**

| Error source | Time optimal | Smooth amplitude | Error type X, Y, Z, LG, AL** |
|---|---|---|---|
| Scattering* $\lvert 1 \rangle$ | 0.103% / 0.043% | 0.036% | 6%, 6%, 25%, 47%, 15% |
| Scattering $\lvert 0 \rangle$ | 0.019% | 0.025% | 7%, 7%, 14%, 62%, 10% |
| Rydberg $T_1 = 88\,\mu s$ | 0.113% | 0.085% | 2%, 2%, 6%, 23%, 67% |
| Rydberg $T_2^* = 3\,\mu s$ | 0.134% | 0.089% | 0%, 0%, 75%, 0%, 25% |
| Position fluct. | 0.012% | 0.054% | 0%, 0%, 96%, 0%, 4% |
| Rydberg $m_{J=-\frac{1}{2}}$ | 0.06 - 0.15% | 0.01% | |
| **Total fidelity** | **99.53 - 99.62%** | **99.70%** | **2%, 2%, 41%, 17%, 38%** |

Simulations were performed at an intermediate-state detuning of Δ/2π=7.8 GHz and a two-photon Rabi frequency Ω/2π=4.6 MHz. *The scattering error for two detuning signs (bright/dark). The total fidelity values assume the correct (dark) detuning choice. **AL, atom loss from population left in the Rydberg state; LG, leakage out of the qubit manifold to other $m_F$ states.