## [Peer Review File · Nature]

Manuscript Title: High-fidelity parallel entangling gates on a neutral atom quantum computer

Reviewer Comments & Author Rebuttals

Reviewer Reports on the Initial Version:

Referees' comments:

Referee #1 (Remarks to the Author):

In the manuscript "High-Fidelity Parallel entangling gates on a neutral atom quantum computer", Evered and authors present ground-breaking new results demonstrating two-qubit gates with a fidelity of 99.5% averaged over arrays of up to 60 qubits to highlight a unique feature of the neutral atom platform in being able to perform operations in an entirely scalable way. Previous work by the team had demonstrated parallel gates on fewer pairs with $F \sim 97.5\%$, and rather than presenting just a minor improvement from increasing the available laser power, the team have achieved these exciting new results by demonstrating a number of novel gate protocols based on using shaped pulse sequences to perform time-optimal and smooth amplitude gates and using a new approach to performing state preparation to minimise the SPAM error. To characterise the gates the team have performed a variety of different measurements based on the principles of randomised benchmarking to carefully evaluate the gate performance, and to my knowledge this is the first demonstration of two-qubit benchmarking with a neutral atom system. Similar analysis is also performed for three-qubit gates, with demonstration of CCZ gates with up to 97.8% again representing the incredible performance the team have achieved in their system.

These results represent a significant step forward in the potential for realising large scale quantum computing architectures, with the performance demonstrated here comparable to the best performance achieved on a chain of 31 ions or 72 superconducting qubits, but with no fundamental limitation in the scalability to more pairs, especially when factoring in the ability to move qubits into the interaction zone using mobile tweezer traps not exploited in this current works. Thus as well as being a major step forward in the neutral atom community, this will impact the wider global community interested in exploiting scalable digital hardware and I therefore strongly recommend this paper be accepted for publication in Nature where it readily meets the requirements for impact and novelty.

The paper itself is incredibly well written, with a clear introduction suitable for a broad audience whilst also providing significant technical details throughout the paper to clearly explain both the methods used to analyse the gates, and clear discussion of the methodology for designing the optimal gate sequences along with subtle details such as the importance of the choice of one photon and two photon detunings to suppress scattering from the intermediate state by engineering the dark state. A thorough error budget is presented along with explanations of routes to further pushing gates to the regime of $F > 0.999$ to make the system capable of reaching future fault tolerant performance. I was also particularly interested in their analysis of correlations between the sites, with this again being the first time sufficiently large ensembles have been used combined with

repeated gate applications to really show how errors and correlations build in the system.

The thorough presentation means I would also like to recommend publication in current form with no changes, but perhaps simply a comment for the authors to consider. For the time optimal gate intermediate state scattering and the coupling to the Rydberg $m_j = -1/2$ state could both be suppressed or eliminated by performing the gate via the $6P_{1/2}$ rather than $6P_{3/2}$ intermediate state by reducing the number of hyperfine states and it no longer being possible to excite both the $+1/2$ and $-1/2$ state if using σ^- and σ^+ polarisations starting in the clock state only $m_j = +1/2$ can be coupled. Is the drawback to this approach simply associated with the challenges of getting necessary laser power on this transition?

Referee #2 (Remarks to the Author):

In their manuscript, Evered et al. demonstrate the realization of two-qubit gates with 99.5% fidelity on up to 60 qubits in parallel using neutral atom arrays. For that, they implement two variants of a control-Z Rydberg blockade gate inspired on the time optimal gate protocol described in Ref [8] by Jandura & Pupillo. This protocol has the advantage of populating atomic dark states which do not contain population in the intermediate state used for the two-photon excitation to the Rydberg state, thereby reducing spontaneous emission. They demonstrate the functionality of these gates by preparing and measuring Bell states in parallel with up to 10 atom pairs. Combined with technical improvements including a higher laser power (offering higher Rabi frequencies at larger intermediate state detunings) and colder atom temperatures, this novel protocol allows them to set a new record for the entangling gate fidelity on stable qubits, now surpassing the threshold for error correction. The gates fidelities extracted through different benchmarks using repetitive measurements are well reproduced and are consistent with the measured Bell fidelities for different system sizes. The contributions to the remaining sources of errors are carefully analyzed and ways for improvements are discussed. Finally, they implement fast multiqubit gates by extending the gate protocol to three qubits, further showing the potential of the platform for practical quantum computing.

Neutral atom platforms offer several advantages over more established architectures for quantum computing. While some of these aspects, such as the scalability in the number of qubits, fast operations, and non-local connectivity have been successfully demonstrated, the two-qubit fidelity was lagging behind. Pushing the 2Q gate fidelity to 99.5% on 60 qubits represents a milestone for tweezer arrays, making it on par with the most advanced architectures. Strikingly, this fidelity does not require recalibrations on individual sites, something which is not common in other platforms. Not only the results are impressive, but the potential for further improvements with realistic technical advances is overwhelming. In my opinion, this work, together with recent results demonstrating mid-circuit erasure conversion with alkaline-earth atom arrays, place Rydberg atom arrays ahead other approaches in the quest for fault-tolerant quantum computing.

The results are very timely and interesting for a broad community. The data contained in the manuscript is of very high quality. It was measured and analyzed using validated protocols. In particular, they follow the approach of Ref [5] by the same group. The consistency in the cross

validation of the extracted Bell state and gate fidelities using different benchmarks is remarkable. Besides, the manuscript is very well written and understandable, even for non-specialists. Previous work is properly acknowledged. The Methods section contains all the relevant information to reproduce their findings. For all the above reasons, I strongly believe that these results deserve the broadest dissemination and therefore I strongly support publication in Nature.

I only have a few questions/comments that I would like the authors to consider:

1. In previous work the authors implemented Raman sideband cooling in their setup. However, in this manuscript, they use grey molasses cooling instead. What is the advantage of using grey molasses? Do they get as well an improved filling fraction of their arrays. I would appreciate a few words explaining their choice.
2. They state that in their setup it is important to keep the Rydberg excitation beams well centered at the positions of the atoms. The beams are shaped into a flat top, so excitation should be insensitive to the positions as long as the intensity is constant over the array. What is the physical reason for that? How do they control the position, using the SLM itself? Along this line, they use a camera to stabilize the position of the Rydberg beams, but they still recalibrate the beam positions based on light shifts measured on the atoms several times per day. Why is it not possible to correlate the position of the beam in the camera and the real beam position as experienced by the atoms? How long does this recalibration procedure take?
3. They use a NA=0.65 objective to allow for separations between the atoms of 2 μm . What is the uncertainty in the position and how does it translate to the interaction energy for the CCZ gates? Does it require to use special algorithms for the holographic generation of the traps?
4. In the Extended Data Fig. 1, they mention that there is a magic wavelength 1 GHz detuned of $6P_{3/2}$ for the $|1\rangle 53S_{1/2}$ transition. Could the authors quantify the 1013 nm light shift with respect to the one obtained with a different sign of the detuning?
5. Also in the Extended Data Fig. 1, the 1013 nm excitation light is on before and after the Raman pulses. What is the reason for that?
6. For the two-qubit gate they use optimal control to obtain a high fidelity. In previous work they use automatic close loops to optimize the performance of sweeps for analog simulation (using the remote dressed chopped-random basis algorithm RedCRAB). Is the empirical optimization of the two-qubit gate fidelity, as illustrated in the Extended Data Fig. 6, performed in this way?
7. In the "Projecting path to 99.9%, and error breakdown" section in the supplementary material, they mention the possibility to convert errors into erasures. This protocol has recently been demonstrated with metastable qubits in alkaline-earth atoms, which seems easier to implement than in Rb. What are the prospects for using this technique in their setup?

Referee responses and summary of revisions

Nature Manuscript number: 2023-04-06004 Evered

June 25, 2023

We would like to thank all Referees for their thorough reading of our manuscript, helpful questions, and positive evaluations. In what follows we address comments point by point and outline minor changes made.

Referee: 1

In the manuscript “High-Fidelity Parallel entangling gates on a neutral atom quantum computer”, Evered and authors present ground-breaking new results demonstrating two-qubit gates with a fidelity of 99.5% averaged over arrays of up to 60 qubits to highlight a unique feature of the neutral atom platform in being able to perform operations in an entirely scalable way. Previous work by the team had demonstrated parallel gates on fewer pairs with $F \sim 97.5\%$, and rather than presenting just a minor improvement from increasing the available laser power, the team have achieved these exciting new results by demonstrating a number of novel gate protocols based on using shaped pulse sequences to perform time-optimal and smooth amplitude gates and using a new approach to performing state preparation to minimise the SPAM error. To characterise the gates the team have performed a variety of different measurements based on the principles of randomised benchmarking to carefully evaluate the gate performance, and to my knowledge this is the first demonstration of two-qubit benchmarking with a neutral atom system. Similar analysis is also performed for three-qubit gates, with demonstration of CCZ gates with up to 97.8% again representing the incredible performance the team have achieved in their system.

These results represent a significant step forward in the potential for realising large scale quantum computing architectures, with the performance demonstrated here comparable to the best performance achieved on a chain of 31 ions or 72 superconducting qubits, but with no fundamental limitation in the scalability to more pairs, especially when factoring in the ability to move qubits into the interaction zone using mobile tweezer traps not exploited in this current work. Thus as well as being a major step forward in the neutral atom community, this will impact the wider global community interested in exploiting scalable digital hardware and I therefore strongly recommend this paper be accepted for publication in Nature where it readily meets the requirements for impact and novelty.

The paper itself is incredibly well written, with a clear introduction suitable for a broad audience whilst also providing significant technical details throughout the paper to clearly explain both the methods used to analyse the gates, and clear discussion of the methodology for designing the optimal gate sequences along with subtle details such as the importance of the choice of one photon and two photon detunings to suppress scattering from the intermediate state by engineering the dark state. A thorough error budget is presented along with explanations of routes to further pushing gates to the regime of $F > 0.999$ to make the system capable of reaching future fault tolerant performance. I was also particularly interested in their analysis of correlations between the sites, with this again being the first time sufficiently large ensembles have been used combined with repeated gate applications to really show how errors and correlations build in the system.

We thank the Referee for their careful reading and positive evaluation of our manuscript.

The thorough presentation means I would also like to recommend publication in current form with no changes, but perhaps simply a comment for the authors to consider. For the time optimal gate intermediate state scattering and the coupling to the Rydberg $m_j = -1/2$ state could both be suppressed or eliminated by performing the gate via the $6P_{1/2}$ rather than $6P_{3/2}$ intermediate state by reducing the number of hyperfine states and it no longer being possible to excite both the $+$ and $-1/2$ state if using sigma- and sigma+ polarisations starting in the clock state only $m_j = +1/2$ can be coupled. Is the drawback to this approach simply associated with the challenges of getting necessary laser power on this transition?

We thank the Referee for this useful comment. We agree that performing the gate via the $6P_{1/2}$ excited state could have certain advantages, in particular to eliminate coupling to the other Rydberg m_J state. As the Referee points out, the main challenge preventing us from utilizing the $6P_{1/2}$ excitation pathway is the smaller available laser power due to a limitation of the existing fiber amplifier technology at this shorter wavelength from $6P_{1/2}$ to the Rydberg state. Moreover, the matrix element to the Rydberg state through $6P_{1/2}$ is smaller than through $6P_{3/2}$.

Based on the Referee's input, we added to the discussion for how to suppress coupling to the other Rydberg m_J state in the Methods section:

To reach 99.9% fidelity, the sum of the errors in Extended Data Fig. 4 needs to be suppressed to below 0.1%, which can be achieved for

example by going 2x further detuned, having a 3x longer Rydberg lifetime (for instance, exciting to a higher n state), 2x longer T_2^* (note that dephasing error scales as $\propto 1/(\Omega T_2^*)^2$), and suppressing coupling to the other m_J state. This suppression can be achieved by applying a larger magnetic field, using the smooth-amplitude gate, or eliminating coupling altogether by exciting from a stretched state or through the $6P_{1/2}$ excited state. An alternative approach to reaching 99.9% fidelity could be going to 3x higher Rabi frequency (again while suppressing coupling to the other m_J state) and 2x larger detuning.

Referee: 2

In their manuscript, Evered et al. demonstrate the realization of two-qubit gates with 99.5% fidelity on up to 60 qubits in parallel using neutral atom arrays. For that, they implement two variants of a control-Z Rydberg blockade gate inspired on the time optimal gate protocol described in Ref [8] by Jandura & Pupillo. This protocol has the advantage of populating atomic dark states which do not contain population in the intermediate state used for the two-photon excitation to the Rydberg state, thereby reducing spontaneous emission. They demonstrate the functionality of these gates by preparing and measuring Bell states in parallel with up to 10 atom pairs. Combined with technical improvements including a higher laser power (offering higher Rabi frequencies at larger intermediate state detunings) and colder atom temperatures, this novel protocol allows them to set a new record for the entangling gate fidelity on stable qubits, now surpassing the threshold for error correction. The gates fidelities extracted through different benchmarks using repetitive measurements are well reproduced and are consistent with the measured Bell fidelities for different system sizes. The contributions to the remaining sources of errors are carefully analyzed and ways for improvements are discussed. Finally, they implement fast multiqubit gates by extending the gate protocol to three qubits, further showing the potential of the platform for practical quantum computing.

Neutral atom platforms offer several advantages over more established architectures for quantum computing. While some of these aspects, such as the scalability in the number of qubits, fast operations, and non-local connectivity have been successfully demonstrated, the two-qubit fidelity was lagging behind. Pushing the 2Q gate fidelity to 99.5% on 60 qubits represents a milestone for tweezer arrays, making it on par with the most advanced architectures. Strikingly, this fidelity does not require recalibrations on individual sites, something which is not common in other platforms. Not only the results are impressive, but the potential for further improvements with realistic technical advances is overwhelming. In my opinion, this work, together with recent results demonstrating mid-circuit erasure conversion with alkaline-earth atom arrays, place Rydberg atom arrays ahead other approaches in the quest for fault-tolerant quantum computing.

The results are very timely and interesting for a broad community. The data contained in the manuscript is of very high quality. It was measured and analyzed using validated protocols. In particular, they follow the approach of Ref [5] by the same group. The consistency in the cross validation of the extracted Bell state and gate fidelities using different benchmarks is remarkable. Besides, the manuscript is very well written and understandable, even for non-specialists. Previous work is properly acknowledged. The Methods section contains all the relevant information to reproduce their findings. For all the above reasons, I strongly believe that these results deserve the broadest dissemination and therefore I strongly support publication in Nature.

We thank the Referee for their positive evaluation of the manuscript.

I only have a few questions/comments that I would like the authors to consider:

1. In previous work the authors implemented Raman sideband cooling in their setup. However, in this manuscript, they use grey molasses cooling instead. What is the advantage of using grey molasses? Do they get as well an improved filling fraction of their arrays. I would appreciate a few words explaining their choice.

Raman sideband cooling has been implemented previously in our research group [Thompson 2013], although in a different experimental setup. We chose to utilize Λ -enhanced gray molasses cooling, primarily because of the experimental simplicity (the 795-nm gray molasses light can be combined with our existing 780-nm PGC path with a single dichroic), as well as the potential to use the same light for enhanced loading. However, for the present work, we do not utilize this enhanced loading technique, as this would slow down our cycle rate, and the initial loading probability does not currently limit our rearrangement filling probability of $\sim 99.5\%$. We added a sentence to the Methods section to clarify:

Subsequently, the atoms are cooled first with polarization gradient cooling (PGC) on the 780-nm D2 line, then with Λ -enhanced gray molasses cooling on the 795-nm D1 line [Grier 2013, Brown 2019, Rosi 2018, Covey 2021]. We implement Λ -enhanced gray molasses cooling due to experimental simplicity (simply combined with our existing PGC path) as well as the potential for enhanced loading [Brown 2019] (which we do not utilize in this work as it reduces our cycle rate).

2. They state that in their setup it is important to keep the Rydberg excitation beams well centered at the positions of the atoms. The beams are shaped into a flat top, so excitation should be insensitive to the positions as long as the intensity is constant over the array. What is the physical reason for that? How do they control the position, using the SLM itself? Along this line, they use a camera to stabilize the position of the Rydberg beams, but they still recalibrate the beam positions based on light shifts measured on the atoms several times per day. Why is it not possible to correlate the position of the beam in the camera and the real beam position as experienced by the atoms? How long does this recalibration procedure take?

We thank the Referee for these clarifying questions. To be maximally insensitive to drift of the beams (even though the beam center is a flat intensity profile), we ensure that the Rydberg beams are well-centered on the atoms. Using our image plane reference camera, the beam positions are actively stabilized using motorized mirror mounts in the Fourier plane. While this reference camera captures most drift of the Rydberg beams relative to the atom array, there is still some drift of the array that is not captured by the camera, in particular primarily coming from a drift in our trapping microscope objective's position which causes the array itself to move relative to the Rydberg beams. To compensate for the relative drift, every few hours we recalibrate the camera reference using the atomic signal, which takes ~ 5 minutes.

To clarify this procedure, we made the following adjustments in the Methods:

We stabilize the beam positions using a reference camera and motorized mirror mounts. To compensate for relative drift between the beam position and the atom array, we re-calibrate the position often (multiple times per day) by stepping the beam positions to maximize the intensity at the atoms as measured by the differential light shift on the qubit transition, which takes ~ 5 minutes. We find that keeping the beams well-centered on the atoms is important to ensure homogeneity and reduce sensitivity to relative beam drifts, and further find that gate parameters are highly reproducible (consistently reproducing fidelities of 99.5%) as long as the beams are properly positioned.

3. They use a NA=0.65 objective to allow for separations between the atoms of $2\ \mu\text{m}$. What is the uncertainty in the position and how does it translate to the interaction energy for the CCZ gates? Does it require to use special algorithms for the holographic generation of the traps?

We thank the Referee for this helpful question. While there can be small uncertainties in the trap positions (e.g. from optical aberrations displacing the center of the trap), the dominant source of positional uncertainty likely comes from the atom positions within the tweezers. In particular, for our atomic temperatures, positional fluctuations can be $\sim 100\text{nm}$, which can lead to $\sim 30\%$ reduction in blockade energy. However, even for this scale of positional fluctuations, the blockade energy is significantly larger than our two-photon Rabi frequency, so we find that positional fluctuations contribute minimally to our CZ gate error budget as shown in Extended Data Fig. 4. A similar picture applies for the CCZ gate, although with certain subtleties for example the $\sqrt{3}$ enhancement in the Rabi frequency for 3 atoms.

We do not use any special algorithms for the holographic generation of the traps, beyond the weighted Gerchberg–Saxton (WGS) algorithm previously used in our group and other groups.

4. In the Extended Data Fig. 1, they mention that there is a magic wavelength 1 GHz detuned of $6P_{3/2}$ for the $|1\rangle \rightarrow 53S_{1/2}$ transition. Could the authors quantify the 1013 nm light shift with respect to the one obtained with a different sign of the detuning?

We thank the Referee for their question. Indeed, there exists a magic wavelength, which is around 1 GHz red-detuned of $6P_{3/2}$ [Lampen 2018], for which there is zero light-shift on the $|1\rangle \rightarrow 53S_{1/2}$ transition coming from the 1013-nm laser. At our experimental detuning of 7.8 GHz red-detuned of $6P_{3/2}$, we operate far away from this magic wavelength condition, such that there is a large 20 MHz light shift. In contrast, if we operated at 7.8 GHz blue-detuned of $6P_{3/2}$, then we estimate that the light shift would be roughly ~ 26 MHz (30% higher), although we have not measured this value experimentally. We made the following change to the caption of Extended Data Fig. 1a:

[...] finally, the 1013-nm light shift is lower (by $\sim 30\%$) at this single-photon detuning sign since there is a magic wavelength ~ 1 GHz red-detuned of $6P_{3/2}$ for the $|1\rangle \rightarrow 53S_{1/2}$ transition [Lampen 2018].

5. Also in the Extended Data Fig. 1, the 1013 nm excitation light is on before and after the Raman pulses. What is the reason for that?

We thank the Referee for this clarifying question. There is no fundamental reason why we need to turn on 1013-nm excitation light for the entire duration of the Bell state generation circuit. We only need to allow enough time for the 1013-nm laser intensity lock to engage before the 420-nm gate pulse is applied, and we found that leaving the laser on for the entire circuit was simple experimentally.

6. For the two-qubit gate they use optimal control to obtain a high fidelity. In previous work they use automatic close loops to optimize the performance of sweeps for analog simulation (using the remote dressed chopped-random basis algorithm RedCRAB). Is the empirical optimization of the two-qubit gate fidelity, as illustrated in the Extended Data Fig. 6, performed in this way?

We thank the Referee for this question. In this work we optimize gate parameters through much simpler parabolic fits. In particular, we start from an analytical gate phase profile with parameters obtained through numerical search

(simple gradient methods), then experimentally scan these gate parameters until we empirically find an optimal set of gate parameters. In future work, closed-loop optimization methods could be employed to automate the search for gate parameters, however in this work we found that manually performing these simple parameter scans was sufficient to reach high gate performance with reduced experimental complexity.

7. In the “Projecting path to 99.9%, and error breakdown” section in the supplementary material, they mention the possibility to convert errors into erasures. This protocol has recently been demonstrated with metastable qubits in alkaline-earth atoms, which seems easier to implement than in Rb. What are the prospects for using this technique in their setup?

We thank the Referee for this useful question. Indeed, owing to the metastable qubit structure in alkaline-earth(-like) atoms, there is a natural way to convert many gate errors into erasure errors [Wu 2022]. For alkali atoms such as rubidium, while there are methods to detect and correct for loss [Cong 2022], we agree that these atoms are less naturally amenable to erasure conversion, although state-selective detection of population in m_F levels outside of the clock qubit subspace could in principle allow for some degree of erasure conversion.

In the main text, we added two references to recent results [Scholl 2023, Ma 2023]. In the Methods we added additional discussion:

Such an understanding is particularly important for quantum error correction [Cong 2022, Wu 2022], for which neutral atoms have various unique opportunities [Singh 2022, Sahay 2023], as knowing the noise structure can be used to enhance the performance of error-correcting schemes. Our present modeling suggests that a majority of errors are Z-type and loss/leakage-type errors, as previously highlighted in Ref. [Cong 2022]. If atom loss is directly detected, these errors would constitute a so-called erasure error [Wu 2022], and moreover, atom loss in this case is in fact a biased erasure error since almost all of it originates from state $|1\rangle$, as pointed out and developed in Ref. [Sahay 2023]. Alkaline-earth(-like) atoms are particularly well-suited to erasure conversion, owing to their metastable qubit structure [Wu 2022, Scholl 2023, Ma 2023].

We again thank the Referee for their positive evaluation, careful reading of the

manuscript, and helpful clarifying questions and comments.